# SimulPL: Aligning Human Preferences in Simultaneous Machine Translation

**Donglei Yu**[1 2]**, Yang Zhao** [1 2]**, Jie Zhu**[3]**, Yangyifan Xu**[1 2]**, Yu Zhou**[1 2]*\* **Chengqing Zong**[1 2]

[1] School of Artificial Intelligence, University of Chinese Academy of Sciences
[2] State Key Laboratory of Multimodal Artificial Intelligence Systems,
Institute of Automation, Chinese Academy of Sciences, Beijing, China
[3] Graduate School of Translation and Interpretation, Beijing Foreign Studies University
{yudonglei2021,zhaoyang2015}@ia.ac.cn
jojo-josephine@bfsu.edu.cn
{yangyifanxu2021,yu.zhou,chengqing.zong}@ia.ac.cn

## Abstract

Simultaneous Machine Translation (SiMT) generates translations while receiving streaming source inputs. This requires the SiMT model to learn a read/write policy, deciding when to translate and when to wait for more source input. Numerous linguistic studies indicate that audiences in SiMT scenarios have distinct preferences, such as accurate translations, simpler syntax, and no unnecessary latency. Aligning SiMT models with these human preferences is crucial to improve their performances. However, this issue still remains unexplored. Additionally, preference optimization for SiMT task is also challenging. Existing methods focus solely on optimizing the generated responses, ignoring human preferences related to latency and the optimization of read/write policy during the preference optimization phase. To address these challenges, we propose Simultaneous Preference Learning (SimulPL), a preference learning framework tailored for the SiMT task. In the SimulPL framework, we categorize SiMT human preferences into five aspects: **translation quality preference**, **monotonicity preference**, **key point preference**, **simplicity preference**, and **latency preference**. By leveraging the first four preferences, we construct human preference prompts to efficiently guide GPT-4/4o in generating preference data for the SiMT task. In the preference optimization phase, SimulPL integrates **latency preference** into the optimization objective and enables SiMT models to improve the read/write policy, thereby aligning with human preferences more effectively. Experimental results indicate that SimulPL exhibits better alignment with human preferences across all latency levels in Zh→En, De→En and En→Zh SiMT tasks. Our data and code will be available at https://github.com/EurekaForNLP/SimulPL.

## 1 Introduction

Simultaneous Machine Translation (SiMT) (Grissom II et al., 2014; Gu et al., 2017; Ma et al., 2019) generates translations while receiving the streaming source inputs. Therefore, the SiMT model needs to learn not only the translation ability but also a read/write policy during training to decide whether to wait for the next incoming source token (READ) or to generate a new target token (WRITE) (Grissom II et al., 2014; Alinejad et al., 2021).

The real-time nature of SiMT scenarios leads to unique human preferences from audiences, which has been demonstrated by relevant linguistic studies (Kurz, 2001; Zwischenberger, 2010). On one hand, the audiences prefer translations that are accurate and easy to understand (Moser, 1996; Sridhar et al., 2013; Dayter, 2020); on the other hand, they also prefer translations to be delivered without unnecessary latency. Fulfilling these preferences is an important goal for interpreters (Amini et al., 2013; Kurz, 2001) and should also be considered in SiMT. However, how to make SiMT models

---

\*Corresponding author.

align with human preferences remains unexplored. Existing SiMT methods (Ma et al., 2019; Alinejad et al., 2021) are primarily trained and evaluated on corpora from the normal offline machine translation (OMT) task, which do not reflect real SiMT scenarios. Some studies (Chen et al., 2020; Guo et al., 2023) have proposed constructing monotonic references to avoid hallucinations, but they still fail to comprehensively consider human preferences.

Furthermore, aligning preferences in the SiMT task presents its own challenges. Existing preference alignment methods (Rafailov et al., 2024; Xu et al., 2024a; Ethayarajh et al., 2024) are designed for tasks such as OMT and question answering, which focus solely on optimizing the model's generated responses. In contrast, these methods have limitations in the SiMT context: they do not account for human preferences regarding latency in the SiMT task and fail to consider enhancing the read/write policy of SiMT models during the preference optimization phase. As a result, these current preference alignment methods are unsuitable for the SiMT task.

To address these issues, we propose Simultaneous Preference Learning (SimulPL), a preference learning framework tailored for the SiMT task. In the SimulPL framework, based on existing research in linguistics and computational linguistics (Moser, 1996; Zwischenberger, 2010; He et al., 2016; Cho, 2016; Chen et al., 2020; Guo et al., 2023), we categorize human preferences in SiMT scenarios and focus on five aspects: **translation quality preference**, **monotonicity preference**, **key point preference**, **simplicity preference**, and **latency preference**. Based on the first four preferences, SimulPL constructs human preference prompts to effectively guide GPT-4/4o in generating preference data for the SiMT task. During the fine-tuning phase, SimulPL proposes Multi-task Supervised Fine-tuning (MSFT) to jointly train the translation ability and read/write policy of the SiMT model for initial preference alignment. Subsequently, SimulPL employs SimulDPO for further preference optimization. During the SimulDPO phase, SimulPL integrates **latency preference** into the optimization objective and enables the SiMT model to further adjust its read/write policy, thereby facilitating more effective alignment with human preferences. We evaluate SimulPL on test sets with references that we manually revised to align with human preferences. Experimental results demonstrate that SimulPL achieves higher translation quality across all latency levels. Furthermore, our manual assessment and multi-aspect evaluation indicate that SimulPL exhibits better alignment with human preferences from both the overall perspective and across the categorized five aspects.

To the best of our knowledge, SimulPL is the first preference learning framework for simultaneous tasks like SiMT. Our contributions can be summarized as follows:

- Our work addresses a critical gap in the study of human preferences for SiMT scenarios. We categorize SiMT human preferences into five aspects: translation quality, monotonicity, key points, simplicity, and latency. This categorization enables the construction of human preference prompts to efficiently guide LLMs in generating preference data for SiMT.

- We propose SimulPL, a preference learning framework tailored for SiMT scenarios. Unlike existing preference learning methods, SimulPL integrates latency preference into the optimization objective and allows the SiMT model to improve its read/write policy during the preference optimization process, enabling better alignment with human preferences.

- Experimental results demonstrate that SimulPL effectively enhances the translation quality across various latency levels. Furthermore, our preference evaluation indicates that SimulPL exhibits better alignment with human preferences.

## 2 RELATED WORK

**Simultaneous Translation** Various SiMT methods introduce different read/write policies. Some approaches propose rule-based fixed policies (Ma et al., 2019; Elbayad et al., 2020), while others focus on adaptive policies that adjust dynamically based on the context. These adaptive policies are modeled in various forms, such as multi-head monotonic attention Ma et al. (2020b), Transducer (Liu et al., 2021), information transport model (Zhang & Feng, 2022), Hidden Markov model (Zhang & Feng, 2023), and self-modifying process (Yu et al., 2024). More recently, some studies (Wang et al., 2023a; Agostinelli et al., 2024; Wang et al., 2024) have also demonstrated the promising performance of large language models in SiMT tasks. However, these efforts are predominantly validated on OMT datasets. Chen et al. (2020) constructed monotonic pseudo-references to reduce unnecessary reorderings. Wang et al. (2023b) generated monotonic references with two-stage beam

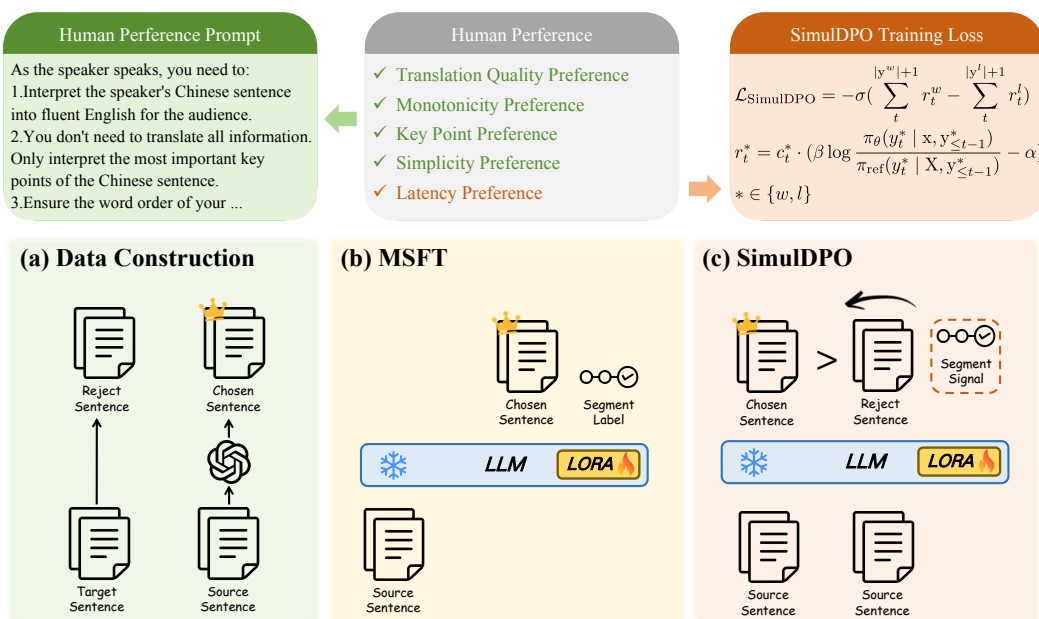

Figure 1: Overview of our proposed SimulPL Framework. With the first four preferences, we construct the human preference prompts to guide GPT-4/4o generating human-preferred translations. The latency preference is integrated into the preference optimization process.

search. Guo et al. (2023) employed RL to balance monotonicity and quality of translations. However, existing work fails to account for real SiMT scenarios and alignment with human preferences.

**LLM Alignment** Aligning LLMs with human preference has become a crucial research challenge recently. Reinforcement Learning from Human Feedback (RLHF) is one of the key approaches (Ouyang et al., 2022; Bai et al., 2022; Yuan et al., 2023). For stable training and less costs, Rafailov et al. (2024) proposed Direct Preference Optimization (DPO), which directly optimizes LLMs without relying on a reward model. Similarly, methods such as CPO (Xu et al., 2024a) and KTO (Ethayarajh et al., 2024) were introduced to improve DPO. Besides, preference alignment is also widely applied to enhance specific tasks (Stiennon et al., 2020; Chen et al., 2024b; Yang et al., 2024). Xu et al. (2024b) explored using RLHF to improve the translation quality. He et al. (2024) utilized the automated evaluation metrics as feedback to enhance translation performance. Nevertheless, existing methods neglect latency preference in the SiMT task and do not improve the read/write policy in the optimization process, both of which negatively impact the alignment in the SiMT task.

## 3 PRELIMINARIES

**Reward Modeling** Existing preference alignment methods typically involve reward modeling and preference optimization. For reward modeling, a human-annotated preference dataset $(x, y^w, y^l)$ is first constructed, where $x$ represents the input, $y^w$ is preferred over $y^l$, which is denoted as $y^w \succ y^l$. Subsequently, existing methods (Christiano et al., 2017; Kim et al., 2023) often train a reward model based on the Bradley-Terry model (Bradley & Terry, 1952), which is formulated as:

$$p(y^w \succ y^l \mid x) = \frac{\exp(r(x, y^w))}{\exp(r(x, y^w)) + \exp(r(x, y^l))} = \sigma(\exp(r(x, y^w)) - \exp(r(x, y^l))) \quad (1)$$

where $r(x, y^w)$ is the score estimated by the reward model, and $\sigma(\cdot)$ is the logistic sigmoid function.

**Preference Optimization** Reinforcement learning (RL) is widely used for preference optimization. Using signals from a reward model, the LLM can be optimized with the following objective:

$$\max_{\pi_\theta} \mathbb{E}_{x \sim D, y \sim \pi_\theta(y|x)}[r(x, y)] - \beta \mathbb{D}_{KL}[\pi_\theta(y \mid x) || \pi_{ref}(y \mid x)] \quad (2)$$

Table 1: Statics of our constructed datasets. We present the reference-free COMET scores of our annotated target sentences with GPT-4/4o and the original target sentences.

| Dataset | Size | | Ref-free COMET | |
|---------|------|------|---------|--------|
| | train | test | GPT-4/4o | Origin |
| Zh→En | 13,491 | 2,000 | 79.13 | 73.72 |
| De→En | 15,717 | 2,168 | 78.93 | 75.02 |
| En→Zh | 19,967 | 2,841 | 80.30 | 76.97 |

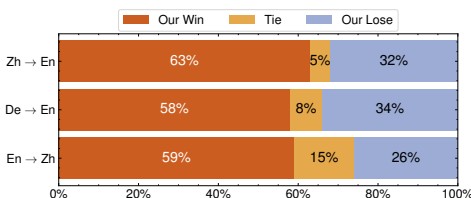

Figure 2: Human evaluation between our annotated target references and origin target references. Our newly annotated references are more preferred.

Additionally, several methods, such as DPO (Rafailov et al., 2024), directly conduct preference alignment without a reward model. However, existing preference alignment methods cannot be directly applied to the SiMT task, as their optimization objectives do not account for the latency preference and do not adjust the read/write policy in the optimization process.

## 4 METHOD: SIMULPL

We propose Simultaneous Preference Learning (SimulPL), a preference learning framework tailored for the SiMT task. The overview of SimulPL is shown in Figure 1. In this framework, we construct human preference prompts based on our categorization of SiMT human preferences to guide GPT-4/4o in generating preference data. During the fine-tuning phase, SimulPL introduces Multi-task Supervised Fine-tuning (MSFT) to jointly learn translation ability and the read/write policy for initial preference alignment. During the preference optimization phase, SimulPL proposes Simultaneous Direct Preference Optimization (SimulDPO), which takes latency preference into account and further improves the read/write policy. The details are discussed in the following.

### 4.1 CATEGORIZATION OF HUMAN PREFERENCE

In real-time SiMT scenarios, the audience exhibits unique human preferences (Kurz, 2001; Zwischenberger, 2010; Amini et al., 2013). Based on existing research in linguistics and computational linguistics, we categorize SiMT human preferences into five aspects:

- **Translation Quality Preference**: Similar to OMT, faithful and fluent translations are also preferred in SiMT (Ma et al., 2019; Miao et al., 2021).

- **Monotonicity Preference**: In the SiMT process, translating monotonically in accordance with the source word order allows for the delivery of translations with minimal pauses (Yang et al., 2023; Chen et al., 2020), which is favored by the audience (Macías, 2006).

- **Key Point Preference**: According to existing research (Moser, 1996; He et al., 2016), concise translations that highlight important information points are more appealing than those that provide complete information in the SiMT scenarios.

- **Simplicity Preference**: In real-time SiMT scenarios, the audience prefers sentences with simpler syntactic structures, which are easier to follow (Sridhar et al., 2013; Dayter, 2020).

- **Latency Preference**: In real-time settings, the audience prefers translations to be delivered without unnecessary latency (Rennert, 2010; Cho, 2016).

It is important to note that latency preference differs from the other four preferences, as it focuses not on the translation content but rather on reducing delays. Therefore, SimulPL aligns with the first four preferences by improving translation ability, and with the latency preference by enhancing the read/write policy.

### 4.2 DATA CONSTRUCTION

**Annotation of Human-preferred Translation**    In our categorization, the first four preferences are reflected in the translation content. Therefore, we utilize them as prior knowledge to construct hu-

man preference prompts and leverage GPT-4/4o (Achiam et al., 2023) to efficiently generate human-preferred translations, denoted as $\mathrm{Y}^w$. The original references, not fully aligned with human preferences, are denoted as $\mathrm{Y}^l$. For the training data, we select subsets from three datasets—WMT15 De→En, WMT22 Zh→En, and MUST-C En→Zh for annotation. The complete prompt used for annotation is provided in Appendix A.1. Correspondingly, newstest2015 De→En, newstest2021 Zh→En, and tst-COMMON are annotated for evaluation. To ensure the accuracy of the test set, we first use GPT-4/4o to generate drafts, and then manually revise them to produce human-preferred references. Our annotators are all qualified in simultaneous interpretation, ensuring reliable and trustworthy revisions. The statistics for our constructed dataset, along with ref-free COMET scores, are shown in Table 1. Notably, we calculate reference-free COMET scores for both kinds of references, showing that our annotated references match the quality of the originals.

To verify that our constructed translation data aligns with human preference, we randomly sample 100 sentences from each of the three language pairs and conducted a manual evaluation by professional simultaneous interpreters. The results in Figure 2 show that our annotated data achieves a higher win rate, indicating stronger alignment with human preference. To further validate our annotated data quality, we conduct additional comparisons between GPT-generated translations and manually revised translations through human evaluation, along with automatic evaluation from the perspective of the first four preferences. These results are available in Appendix A.2.

**Prefix Pairs Extraction**   To enable the SiMT model in learning translation based on source prefixes instead of complete source sentences, we extract prefix pairs from our annotated sentence pairs using word alignment and add them to the training data. For each sentence pair $(\mathrm{X}, \mathrm{Y}^w)$, we use awesome-align (Dou & Neubig, 2021) to get the word alignment. For target token $y_t$, we denote the corresponding source token as $x_{a_t}$, and the set of extracted prefix pairs are denoted as:

$$D_p^w = \{(\mathrm{x}, \mathrm{y}^w) \mid \text{if } 0 < t \leq |\mathrm{y}^w|, \text{then } 0 < a_t \leq |x|; a_{|\mathrm{y}^w|+1} > |x|\} \tag{3}$$

Intuitively, for the given source prefix x, the target prefix $\mathrm{y}^w$ includes all the translatable content. Similarly, we can extract prefix pairs from sentence pairs $(\mathrm{X}, \mathrm{Y}^l)$ to obtain $D_p^l$. Then, we merge $D_p^w$ and $D_p^l$ to create the prefix-level preference dataset:

$$D_p = \{(\mathrm{x}, \mathrm{y}^w, \mathrm{y}^l) \mid (\mathrm{x}, \mathrm{y}^w) \in D_p^w, (\mathrm{x}, \mathrm{y}^l) \in D_p^l\} \tag{4}$$

## 4.3   Multi-task Supervised Fine-tuning

Based on a pre-trained language model $\pi_{\mathrm{pre}}$, SimulPL introduces Multi-task Supervised Fine-tuning (MSFT) to jointly learn translation ability and read/write policy on $D_p^w$ for initial preference alignment. For translation ability, the model learns to generate the target prefix $\mathrm{y}^w$ from the source prefix x. For read/write policy, SimulPL adds an extra confidence layer, consisting of a linear layer and a sigmoid layer, to make read/write decisions. Specifically, when predicting $y_t^w$, an additional confidence $c_t^w$ is estimated by the confidence layer. If $t < |\mathrm{y}^w|$, the model should predict $c_t^w = 1$, indicating the WRITE decision. Otherwise, if $t > |\mathrm{y}^w|$, the model should estimate $c_t^w = 0$, which means it should stop translating and choose the READ decision. The complete training loss for the MSFT phase is calculated as:

$$\mathcal{L}_{\mathrm{MSFT}} = -\prod_{t=1}^{|\mathrm{y}^w|} y_t^w \log \pi_{\mathrm{sft}}(y_t^w \mid \mathrm{x}, \mathrm{y}_{\leq t-1}^w) - \prod_{t=1}^{|\mathrm{y}^w|+1} [\mathbb{I}(t \leq |\mathrm{y}^w|) \log c_t^w + \mathbb{I}(t > |\mathrm{y}^w|) \log(1 - c_t^w)] \tag{5}$$

where $\pi_{\mathrm{sft}}$ is initialized with the parameters of $\pi_{\mathrm{pre}}$, and $\mathbb{I}(\cdot)$ denotes the indicator function. It is noted that we train the model to predict $c_{|\mathrm{y}^w|+1}^w = 0$, allowing the SiMT model to learn to stop translating at the appropriate position.

## 4.4   Simultaneous Direct Preference Optimization

After the MSFT phase, SimulPL introduces Simultaneous Direct Preference Optimization (SimulDPO) to further align with human preferences. In the SimulDPO phase, SimulPL integrates the latency preference into the optimization objective and allows the SiMT model to further improve its read/write policy during preference optimization.

First, we modify the optimization objective from Equation 2 to encourage the SiMT model to generate human-preferred translations while satisfying additional latency preference. Specifically, we add an output length constraint, which can be expressed as follows:

$$\max_{\pi_\theta} \mathbb{E}_{\mathrm{x}\sim D, \mathrm{y}\sim\pi_\theta(\mathrm{y}|\mathrm{x})}[r(\mathrm{x},\mathrm{y})] - \beta\mathbb{D}_{\mathrm{KL}}[\pi_\theta(\mathrm{y}\mid\mathrm{x})||\pi_{\mathrm{ref}}(\mathrm{y}\mid\mathrm{X})] + \alpha\mathbb{E}[|\mathrm{y}|] \qquad (6)$$

where $\alpha$ is a hyper-parameter introduced to control the output length constraint. Unlike $\pi_\theta$ which only accesses the source prefix x, $\pi_{\mathrm{ref}}$ is provided with the complete source sentence X to generate a more accurate prediction, thus effectively preventing $\pi_\theta$ from diverging. Intuitively, based on the received x, the SiMT model is encouraged to translate as much content as possible to minimize unnecessary latency. Theoretically, we prove that the constraint $\mathbb{E}[|\mathrm{y}|]$ aligns with the objective of reducing latency, which is provided in Appendix B.1. Therefore, although similar in form to R-DPO (Park et al., 2024), our goals and methods are entirely opposite, as detailed in Appendix C.

Based on Equation 6, we can derive the following reward function concerning $\pi_\theta$ and $|\mathrm{y}|$:

$$r(\mathrm{x},\mathrm{y}) = \beta\log\frac{\pi_\theta(\mathrm{y}\mid\mathrm{x})}{\pi_{\mathrm{ref}}(\mathrm{y}\mid\mathrm{X})} + \beta\log Z(\mathrm{x}) - \alpha|\mathrm{y}| \qquad (7)$$

where $Z(\mathrm{x})$ is the partition function. The detailed derivation is provided in Appendix B.2.

Since both $\pi_\theta$ and $\pi_{\mathrm{ref}}$ are initialized from $\pi_{\mathrm{sft}}$, they also possess the ability to estimate confidence. We denote the confidence estimated by $\pi_\theta$ when predicting $y_t$ as $c_t$, leading to the following expression: $\pi_\theta(\mathrm{y}\mid\mathrm{x}) = \prod_{t=1}^{|\mathrm{y}|+1}[\pi_\theta(y_t\mid\mathrm{x},\mathrm{y}_{\leq t-1})]^{\mathbb{I}(t\leq|\mathrm{y}|)} = \prod_{t=1}^{|\mathrm{y}|+1}[\pi_\theta(y_t\mid\mathrm{x},\mathrm{y}_{\leq t-1})]^{c_t}$. Intuitively, we can also gain that: $|\mathrm{y}| = \sum_{t=1}^{|\mathrm{y}|+1}\mathbb{I}(t\leq|\mathrm{y}|) = \sum_{t=1}^{|\mathrm{y}|+1}c_t$. Substituting these two equations into Equation 7, we obtain the following representation of $r(\mathrm{x},\mathrm{y})$:

$$r(\mathrm{x},\mathrm{y}) = \beta\sum_{t=1}^{|\mathrm{y}|+1}c_t\log\frac{\pi_\theta(y_t\mid\mathrm{x},\mathrm{y}_{\leq t-1})}{\pi_{\mathrm{ref}}(y_t\mid\mathrm{X},\mathrm{y}_{\leq t-1})} + \beta\log Z(\mathrm{x}) - \alpha\sum_{t=1}^{|\mathrm{y}|+1}c_t \qquad (8)$$

Then, with the Bradley-Terry model, we can derive the training objective of SimulDPO as follows:

$$\mathcal{L}_{\mathrm{SimulDPO}} = -\log\sigma\Big(\sum_{t}^{|\mathrm{y}^w|+1}r_t^w - \sum_{t}^{|\mathrm{y}^l|+1}r_t^l\Big)$$

$$r_t^* = c_t^*\cdot\Big(\beta\log\frac{\pi_\theta(y_t^*\mid\mathrm{x},\mathrm{y}_{\leq t-1}^*)}{\pi_{\mathrm{ref}}(y_t^*\mid\mathrm{X},\mathrm{y}_{\leq t-1}^*)} - \alpha\Big), \quad *\in\{w,l\} \qquad (9)$$

This training loss enables the SiMT model to further improve its read/write policy during the preference alignment phase, and takes the latency preference into account. During training, if the SiMT model can accurately predict a token that aligns with human preferences (i.e., $\log\frac{\pi_\theta(y_t^w|\mathrm{x},\mathrm{y}_{\leq t-1}^w)}{\pi_{\mathrm{ref}}(y_t^w|\mathrm{X},\mathrm{y}_{\leq t-1}^w)} > \frac{\alpha}{\beta}$), then the SiMT model will learn to predict $c_t^w$ close to 1 to avoid latency. Conversely, if the prediction does not align well, $c_t^w$ should be predicted close to 0. Additionally, since the read/write decisions for $\mathrm{y}^l$ are not the focus of the optimization, we directly set $c_t^l = \mathbb{I}(t\leq|\mathrm{y}^l|)$, instead of using the predictions from the SiMT model during training.

## 4.5 CONFIDENCE-BASED POLICY DURING INFERENCE

During inference, SimulPL makes decisions based on estimated $c_t$. If $c_t > 0.5$, it indicates that the SiMT model can generate a target token $y_t$ aligned with human preferences and should choose WRITE; otherwise, it chooses READ. Following Wang et al. (2023a), we introduce the reading length $n$ to control the latency level of SimulPL. Specifically, when the SiMT model chooses READ, it needs to wait for $n$ new source words before making further decisions. In Appendix D, we provide more details with Algorithm 1 and analyze the impact of the confidence threshold.

## 5 EXPERIMENTS

### 5.1 EXPERIMENTAL DETAILS

**Datasets** We validate our method on text-to-text SiMT tasks using our annotated datasets with human-preferred references. For Transformer-based SiMT models, we first pre-train them on the

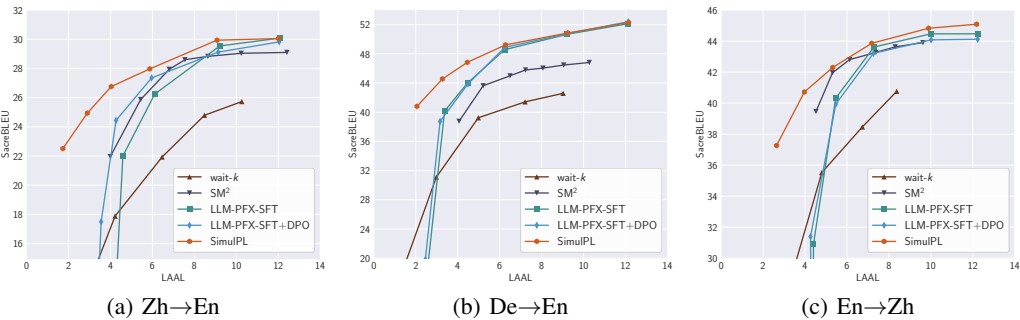

Figure 3: SacreBLEU against LAAL on Zh→En, De→En and En→Zh SiMT tasks.

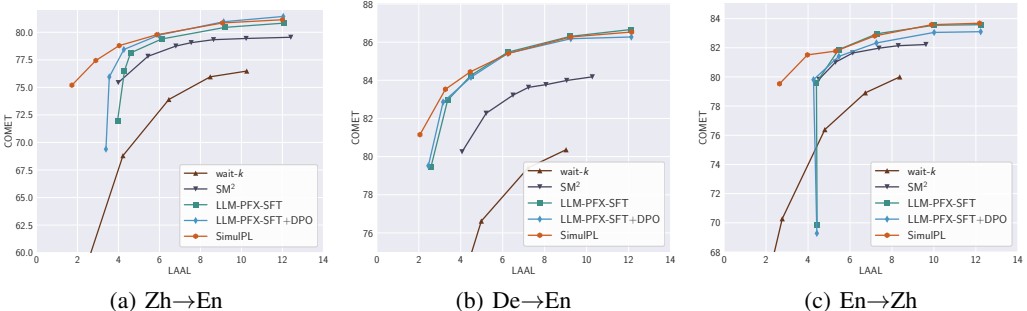

Figure 4: COMET against LAAL on Zh→En, De→En and En→Zh SiMT tasks.

complete OMT training sets and then fine-tune on our annotated data. For LLM-based SiMT models, we begin with SFT on a subset of the OMT training data with the size of 100k, followed by additional SFT on our annotated data, so that LLMs can initially learn the translation ability.

**Baselines** Existing SiMT models primarily include Transformer-based and LLM-based architectures. We reproduce both types of SiMT models, as detailed in the following:

- **wait-$k$** (Ma et al., 2019): This Transformer-based SiMT model first waits for $k$ source tokens. It then repeatedly generates a target token and waits for a source token.
- **SM$^2$** (Yu et al., 2024): This Transformer-based SiMT model learns the confidence of current prediction during training. During inference, it decides whether to wait for an additional source token or output a target token based on the confidence.
- **LLM-PFX-SFT** (Wang et al., 2023a): This LLM-based SiMT model utilizes prefix-pairs data for SFT and utilizes incremental decoding during inference.
- **LLM-PFX-SFT+DPO**: Building on LLM-PFX-SFT, we further align preferences using our constructed prefix-level preference data through a standard DPO method.

**Implementation Details** We implement Transformer-based SiMT models using Fairseq (Ott et al., 2019). For the Zh→En and En→Zh SiMT tasks, we use vocabularies of 45,000 for Chinese and 35,000 for English respectively. The De→En task is applied with a shared vocabulary of 32,000. For LLM-based SiMT models, following, we choose Llama2-7B-chat (Touvron et al., 2023) as the base model and use Simuleval (Ma et al., 2020a) for evaluation. To address vocabulary inconsistencies among SiMT models, we allow word-level read/write operations during testing to facilitate accurate latency comparison. More implementation details are available in Appendix E.

**Evaluation Metrics** We use specific metrics to measure our categorized human preferences respectively. For *Translation Quality Preference*, we utilize SacreBLEU (Post, 2018) and COMET as the metrics. For *Latency Preference*, we chose Length-Adaptive Average Lagging (LAAL) (Papi et al., 2022) to avoid misjudging the over-generation phenomena. A higher LAAL indicates greater latency. For *Monotonicity Preference*, we define Normalized Inversion Rate (NIR) to measure the monotonicity. Specifically, we first use awesome-align to obtain word alignment between the

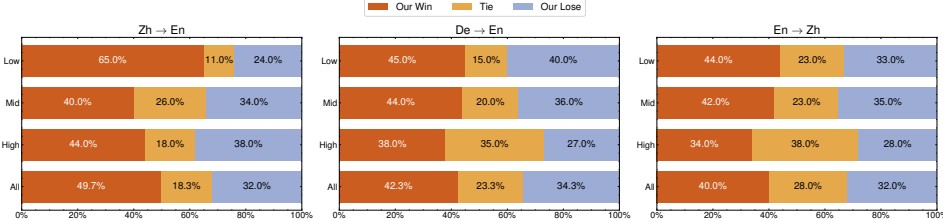

(a) Human evaluation between SimulPL and LLM-PFX-SFT.

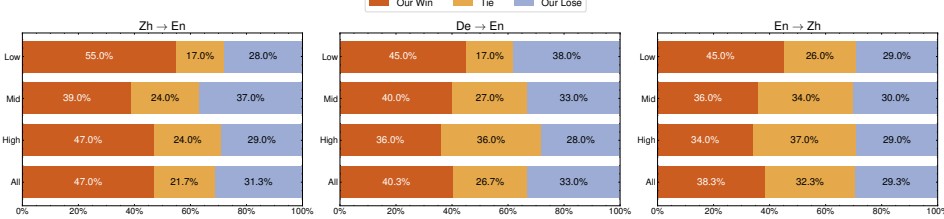

(b) Human evaluation between SimulPL and LLM-PFX-SFT+DPO.

Figure 5: Human Evaluation in differenct latency groups on Zh→En, De→En and En→Zh SiMT tasks. We divide the latency into three levels: low latency ($0 \le$ LAAL $< 4$), medium latency ($4 \le$ LAAL $< 8$), and high latency (LAAL $\ge 8$). We report the independent evaluation results at each latency group and the overall results across all latency groups.

Table 2: Multi-aspect Evaluation in different latency groups on Zh→En, De→En and En→Zh SiMT tasks. We define Normalized Inversion Rate (NIR), Sentence Length Ratio (SLR) and Dependency Depth (DD) for respectively measuring monotonicity preference, key point preference and simplicity preference. COMET and SacreBLEU are used to measure translation quality preference.

| Models | Preference Analysis | | | | | | | | | | | | | | |
|---|---|---|---|---|---|---|---|---|---|---|---|---|---|---|---|
| | Zh→En | | | | | De→En | | | | | En→Zh | | | | |
| | NIR ↓ | DD ↓ | SLR ↓ | COMET | BLEU | NIR ↓ | DD ↓ | SLR ↓ | COMET | BLEU | NIR ↓ | DD ↓ | SLR ↓ | COMET | BLEU |
| *Low Latency* | | | | | | | | | | | | | | | |
| LLM-PFX-SFT | 5.24 | 7.60 | 0.90 | 74.25 | 12.05 | 3.03 | 5.95 | 1.32 | 81.21 | 29.61 | 5.21 | 5.72 | 2.62 | 74.74 | 19.23 |
| LLM-PFX-SFT+DPO | 3.39 | 7.97 | 0.80 | 72.67 | 14.79 | 2.97 | **5.76** | 1.29 | 81.19 | 29.29 | 5.13 | 5.68 | 2.55 | 74.55 | 19.62 |
| SimulPL | **2.95** | **6.97** | **0.64** | **76.32** | **23.73** | **2.69** | 6.02 | **1.01** | **82.34** | **42.69** | **4.81** | **5.46** | **0.96** | **80.52** | **38.99** |
| *medium latency* | | | | | | | | | | | | | | | |
| LLM-PFX-SFT | 4.65 | 7.02 | 0.65 | 78.76 | 24.13 | 3.80 | 5.97 | 1.01 | 84.86 | 46.31 | 7.65 | 5.43 | 0.97 | **82.42** | 41.97 |
| LLM-PFX-SFT+DPO | **3.49** | 7.09 | 0.63 | 79.14 | 25.90 | 3.88 | **5.94** | 1.00 | 84.79 | 46.39 | 7.55 | 5.44 | 0.97 | 81.87 | 41.58 |
| SimulPL | 4.21 | **6.91** | **0.62** | **79.29** | **27.37** | **3.60** | 5.96 | **0.99** | **84.93** | **48.03** | **7.11** | **5.38** | **0.95** | 82.29 | **43.08** |
| *High Latency* | | | | | | | | | | | | | | | |
| LLM-PFX-SFT | 5.66 | 6.82 | 0.61 | 80.64 | 29.81 | 4.53 | 5.93 | 0.98 | **86.48** | 51.41 | 9.48 | 5.40 | 0.91 | 83.56 | 44.47 |
| LLM-PFX-SFT+DPO | **4.60** | 6.85 | 0.62 | **81.25** | 29.47 | 4.70 | **5.90** | 0.97 | 86.23 | **51.60** | 9.34 | 5.37 | **0.91** | 83.08 | 44.10 |
| SimulPL | 5.59 | **6.78** | **0.60** | 81.00 | **30.00** | **4.44** | 5.92 | **0.97** | 86.40 | 51.58 | **9.10** | **5.37** | 0.92 | **83.63** | **44.96** |

model's output $\hat{Y}$ and the source sentence X. We denote the source position corresponding to $\hat{y}_t$ as $\hat{a}_t$. These source positions form a sequence $\hat{A} = [\hat{a}_1, \hat{a}_2, \dots]$. The inversion number (Mannila, 1985) of $\hat{A}$, denoted as $I_{\hat{A}}$, reflects the monotonicity of $I_{\hat{A}}$. A smaller $I_{\hat{A}}$ indicates a more monotonic translation, while a larger $I_{\hat{A}}$ suggests more reordering in the translation. Since $I_{\hat{A}}$ is affected by $|\hat{A}|$, we define NIR $= \frac{2I_{\hat{A}}}{|\hat{A}|(|\hat{A}|-1)} \times 100\%$ as the metric for monotonicity preference. For *Key Point Preference*, we use the Sentence Length Ratio (SLR) as the metric, defined as SLR $= \frac{|\hat{Y}|}{|X|}$. A smaller SLR indicates that the SiMT model completes the translation with a shorter sentence, aligning better with key point preference. For *Simplicity Preference*, we use Stanza (Qi et al., 2020) to convert $|\hat{Y}|$ into a dependency tree $\hat{T}$, and define the depth of $\hat{T}$ as Dependency Depth (DD) for evaluation. A smaller DD indicates simpler syntax, which is easier for audiences to follow.

## 5.2 TRANSLATION QUALITY

The SacreBLEU and COMET scores for different SiMT methods are shown in Figure 3 and Figure 4. These results show that our proposed SimulPL achieves higher translation quality across all latency levels on three language pairs, particularly in low latency level. This indicates that SimulPL better meets the translation quality preference in SiMT scenarios. Since the test set we used includes

human-aligned references, we argue this can also reflect the effectiveness of SimulPL in aligning with other human preferences. We provide detailed numerical results and values for other latency metrics in Appendix F.

## 5.3 PREFERENCE EVALUATION

To validate the effectiveness of SimulPL in preference alignment, we conduct a further human preference evaluation for LLM-PFX-SFT, LLM-PFX-SFT+DPO, and SimulPL. This evaluation includes the overall human evaluation and multi-aspect evaluation. Specifically, we first divide the models' outputs into three latency groups: low latency ($0 \leq \text{LAAL} < 4$), medium latency ($4 \leq \text{LAAL} < 8$), and high latency ($\text{LAAL} \geq 8$). For human evaluation, we manually assess 100 sentences sampled from each latency group. Our evaluators are all qualified in simultaneous interpretation and can provide accurate assessments. For multi-aspect evaluation, we measure the performance of these SiMT models in terms of translation quality preference, monotonicity preference, key point preference, and simplicity preference in different latency groups. Detailed analyses are provided as follows.

**Human Evaluation** The results of the human evaluation are shown in Figure 5, which show that SimulPL achieves higher win rates in all latency groups for these three language pairs. This indicates that SimulPL can generate translations more aligned with human preferences. We attribute this performance improvement to the joint optimization of translation ability and read/write policy during the preference alignment process.

**Multi-aspect Evaluation** The results of muti-aspect evaluation are shown in Table 2. SimulPL achieves better alignment across all latency groups for the three language pairs. SimulPL not only maintains high translation quality but also effectively manages monotonicity, key points, and simpler syntactic structures. Under low latency conditions, SimulPL achieves a better trade-off between latency preference and other preferences and generates better translations.

## 5.4 ABLATION STUDIES

We conduct ablation studies on SimulPL for the Zh→En SiMT task, with detailed analyses in the following. Additional results and other analyses on De→En SiMT task are shown in Appendix G.

**Effect of MSFT** To verify the role of MSFT, we evaluate the performance of a SiMT model trained with regular SFT using prefix pairs data, similar to LLM-PFX-SFT. The results in Figure 6 show that MSFT outperforms SFT, especially in the low latency level. This shows that by explicitly modeling the multi-task of translation ability and read/write policy, MSFT improves the SiMT performance more effectively and provides better initialization parameters for SimulDPO.

**Effect of SimulDPO** As shown in Figure 6, SimulPL, which introduces SimulDPO after MSFT phase, achieves higher SacreBLEU scores across various latency levels compared to Only MSFT. This indicates SimulPL further enhances the translation ability and read/write policy during the SimulDPO phase, leading to better performance.

**Effect of $\pi_{\text{ref}}(\text{y} \mid \text{X})$** To verify whether the predicted $\pi_{\text{ref}}(\text{y} \mid \text{X})$ in OMT setting can provide a more accurate constraint for the training objective, we replace $\pi_{\text{ref}}(\text{y} \mid \text{X})$ in SimulPL with the probability $\pi_{\text{ref}}(\text{y} \mid \text{x})$ and evaluate the performance in this setting. As shown in Figure 6, the performance of SimulPL trained with $\pi_{\text{ref}}(\text{y} \mid \text{x})$ obviously declines. We argue this is due to the inaccurate prediction of $\pi_{\text{ref}}(\text{y} \mid \text{x})$ negatively impacts the preference optimization.

## 5.5 IMPACT OF $\alpha$ ON BALANCING ALIGNMENT AND LATENCY DURING TRAINING

In SimulDPO, $\alpha$ is introduced as a hyper-parameter into the training loss. As shown in Equation 9, $\alpha$ functions as a token-level threshold. Since the gradient of $\alpha c_t^l$ does not propagate, we only analyze the impact of $\alpha c_t^w$. Specifically, during training, if $\log \frac{\pi_\theta(y_t^w|\text{x},\text{y}_{\leq t-1}^w)}{\pi_{\text{ref}}(y_t^w|\text{X},\text{y}_{\leq t-1}^w)} > \frac{\alpha}{\beta}$, we consider that $\pi_\theta(y_t^w \mid \text{x}, \text{y}_{\leq t-1}^w)$ presents a prediction aligning human preferences well, and the SiMT model should learn to predict a higher $c_t^w$. Conversely, the model should learn a lower $c_t^w$ when this condition is not met. Thus, appropriately increasing the value of $\alpha$ can enhance the model's ability to learn better alignment quality. However, if $\alpha$ is set too high, the SiMT model could become overly cautious in translation, leading to $c_t^w$ failing to accurately balance between latency preference

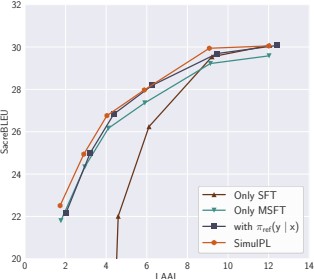 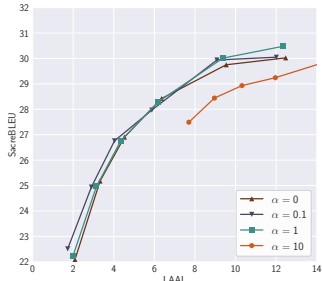 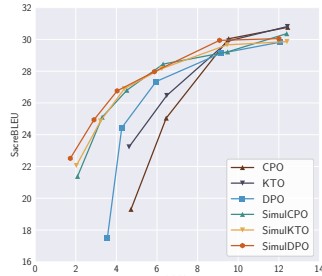

Figure 6: Ablation Studies of SimulPL framework on Zh→En SiMT task.

Figure 7: The comparison of SimulDPO with difference $\alpha$ values on Zh→En SiMT task.

Figure 8: The comparison between different preference optimization methods.

and other preferences. To validate the effect of $\alpha$, we train SimulPL with different $\alpha$ values and compare their performance on Zh→En task. The results are shown in Figure 7. When $\alpha = 10$, the SiMT model always estimates a low confidence and tends to wait for more tokens before generating translation. On the other hand, when $\alpha$ is too small ($\alpha = 0$), the model's performance decreases to some extent. We will further explore the impact of $\alpha$ in future work.

## 5.6 GENERALIZATION TO OTHER PREFERENCE OPTIMIZATION METHODS

In the SimulPL framework, SimulDPO is introduced for further preference optimization, which is adapted from DPO. Theoretically, SimulPL has generalization to any preference optimization methods, making them applicable to the SiMT scenarios. To validate this generality, we integrate CPO and KTO into the SimulPL framework, deriving SimulCPO and SimulKTO. Their respective training losses are provided in the following:

$$\mathcal{L}_{\text{SimulCPO}} = -\log \sigma\Big( \sum_t^{|\mathbf{y}^w|+1} u_t^w - \sum_t^{|\mathbf{y}^l|+1} u_t^l \Big) - \log \pi_\theta(\mathbf{y} \mid \mathbf{x}) \tag{10}$$

$$u_t^* = \beta c_t^* \log \pi_\theta(y_t^* \mid \mathbf{x}, \mathbf{y}_{\leq t-1}^*) - \alpha c_t^*, \quad * \in \{w, l\}$$

$$\mathcal{L}_{\text{SimulKTO}} = \lambda_{\mathbf{y}} - v(\mathbf{x}, \mathbf{y})$$

$$v(\mathbf{x}, \mathbf{y}) = \begin{cases} \lambda_w \sigma(\sum_t^{|\mathbf{y}|+1} r_t^w - z_0), & \text{if } \mathbf{y} \sim \mathbf{y}^w \mid \mathbf{x} \\ \lambda_l \sigma(z_0 - \sum_t^{|\mathbf{y}|+1} r_t^l), & \text{if } \mathbf{y} \sim \mathbf{y}^l \mid \mathbf{x} \end{cases} \tag{11}$$

where $\lambda_{\mathbf{y}}$ denotes $\lambda_w$ ($\lambda_l$) when y is desirable (undesirable), and $z_0 = \mathbb{D}_{\text{KL}}[\pi_\theta(\mathbf{y} \mid \mathbf{x})||\pi_{\text{ref}}(\mathbf{y} \mid \mathbf{X})]$.

We evaluate their performances on the Zh→En SiMT task. As shown in Figure 8, both SimulCPO and SimulKTO achieve higher performance compared to CPO and KTO, particularly in low-latency levels. These results indicate the generalization of SimulPL. Additionally, SimulDPO, SimulCPO, and SimulKTO exhibit similar performance, making it difficult to determine which is most suitable for SiMT task. Besides preference optimization methods, SimulPL may also generalize to other tasks like simultaneous inference (Chen et al., 2024a). We will explore this in future work.

## 6 CONCLUSION

We bridge the gap in the study of SiMT human preferences and propose SimulPL, a preference learning framework tailored for SiMT task. Drawing from existing research, we categorize preferences in SiMT scenarios into five aspects: translation quality, monotonicity, key points, simplicity, and latency. By leveraging the first four preferences, SimulPL constructs human preference prompts to efficiently guide LLMs in generating preference data for SiMT. During the fine-tuning phase, SimulPL introduces MSFT for initial preference alignment. During the preference optimization phase, SimulPL proposes SimulDPO, integrating latency preference into the optimization objective and further improving the read/write policy. Our experiments indicate that SimulPL achieves better preference alignment both overall and across each aspect. Additionally, our analysis shows that SimulPL has a generalization to other preference optimization methods.

## ACKNOWLEDGMENTS

Acknowledgment The research work has been supported by the Natural Science Foundation of China under Grant No. 62336008 and No. 62476271.

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

## A    DATASET CONSTRUCTION AND ANALYSIS

### A.1    HUMAN PREFERENCE PROMPTS

Based on our categorized SiMT human preferences, we construct human preference prompts, which account for translation quality preference, monotonicity preference, key point preference, and simplicity preference, to efficiently guide GPT-4/4o in generating preference data for the SiMT task. Taking Zh→En SiMT task as an example, our complete human preferences prompts are shown in Figure 9.

---

**System**:
You are a professional simultaneous interpreter, simulating the scenario of interpreting a speaker's Chinese speech into English in real time.
**User:**
As the speaker speaks, you need to:
1.Interpret the speaker's Chinese sentence into fluent English for the audience.
2.You don't need to translate all information. Only interpret the most important key points of the Chinese sentence.
3.Ensure the word order of your interpretation must be the same with the word order in the Chinese sentences.
4.Use simple, common words, and keep the syntax very easy. Make your interpretation as short as possible.

Now, I will provide you with two examples.

Input:
上周失去耐克赞助商身份的布朗表示，他将从迈阿密开始，每周在高中训练一天。
Output:
{
    key points order: 1.上周 2.失去耐克赞助商 3.布朗表示 4.他将从迈阿密开始 5.每周在高中训练一天,
    interpretation: Last week, losing his Nike sponsorship, Brown said he would start from Miami and train at a high school one day per week.
}

Input:
这是茂木就任外相后，作为和平条约缔结谈判负责人进行的首次对俄磋商。
Output:
{
key points order: 1.茂木就任外相后 2.作为和平条约缔结谈判负责人 3.首次对俄磋商,
interpretation: After assuming office as Foreign Minister, as the chief negotiator for the peace treaty, Motegi held his first negotiations with Russia.
}

Input:
<source sentence>

---

Figure 9: Our constructed human preferences prompts on Zh→En SiMT task.

### A.2    FURTHER EVALUATION OF OUR ANNOTATED DATASETS.

We conduct both automated muti-aspect evaluation and additional human evaluation to validate the quality of our constructed dataset further. The details are described in the following.

**Multi-aspect Evaluation.**    Similar to Section 5.3, we also use our defined NIR, SLR, and DD to conduct multi-aspect evaluation on the GPT-generated references and the original references. We use Ref-free COMET to assess the translation quality here. The results in Table 3 indicate that GPT-4/4o aligns better with human preferences.

Then, we compare the multi-aspect evaluation results of GPT-generated translations and those manually revised by interpreters on the test sets. To facilitate comparison, we also provide the results for the original references of the test sets. As shown in Table 4, the translations generated by GPT-4/4o are either superior to or comparable with the original references in terms of monotonicity, key points, simplicity, and translation quality. Moreover, these results are very close to the manually revised translations. This indicates that the quality of the GPT-generated data is both reliable and aligned well with human preferences.

Table 3: Multi-aspect evaluation on our annotated references and original references.

| References | Zh→En | | | | De→En | | | | En→Zh | | | |
|---|---|---|---|---|---|---|---|---|---|---|---|---|
| | NIR ↓ | DD ↓ | SLR ↓ | COMET | NIR ↓ | DD ↓ | SLR ↓ | COMET | NIR ↓ | DD ↓ | SLR ↓ | COMET |
| GPT-4/4o | 11.39 | 5.48 | 0.55 | 79.13 | 9.99 | 5.61 | 0.92 | 78.93 | 25.20 | 5.52 | 0.89 | 80.30 |
| Origin | 45.41 | 6.34 | 0.56 | 73.72 | 34.37 | 6.31 | 1.09 | 75.02 | 44.05 | 6.20 | 0.94 | 76.97 |

Table 4: Multi-aspect evaluation on test sets.

| References | Zh→En | | | | De→En | | | | En→Zh | | | |
|---|---|---|---|---|---|---|---|---|---|---|---|---|
| | NIR ↓ | DD ↓ | SLR ↓ | COMET | NIR ↓ | DD ↓ | SLR ↓ | COMET | NIR ↓ | DD ↓ | SLR ↓ | COMET |
| GPT-4/4o | 6.48 | 6.55 | 0.49 | 78.32 | 5.49 | 5.71 | 0.92 | 80.86 | 10.65 | 5.63 | 0.88 | 81.24 |
| Manually Revised | 6.71 | 6.69 | 0.52 | 79.05 | 5.52 | 5.84 | 0.95 | 81.89 | 10.47 | 5.63 | 0.88 | 81.42 |
| Origin | 13.21 | 7.53 | 0.74 | 79.05 | 7.80 | 6.29 | 1.09 | 80.78 | 12.89 | 6.19 | 0.89 | 75.77 |

**Human evaluation.** Following Kocmi et al. (2022) and Xu et al. (2024a), we randomly sample 200 sentences from the test set and employ professional interpreters, who are not involved in the revision process, to manually assess their GPT-generated translations and manually-revised translations. This evaluation is conducted on our test sets. The evaluation criteria are as follows:

- **0:** The translation is poor and fails to convey any meaningful information.
- **2:** The translation conveys some of the speaker's information but misses key points. It also includes unnecessary reordering, leading to unnecessary delays in real-time scenarios, and uses less common expressions that make it harder for the audience to quickly understand.
- **4:** The translation conveys all the important information with minimal unnecessary reordering. It uses simple expressions that generally meet the audience's needs, though there are still some minor issues.
- **6:** The translation is a perfect interpretation, accurately conveying the speaker's key points while omitting unnecessary details. It uses expressions that align with spoken language conventions, significantly reflecting human preferences in simultaneous interpretation.

The results are shown in Table 5. The average scores of GPT-generated translations are close to those are manually revised. Besides, the score distribution shows that most of the GPT-generated translations scored 4 or higher. These results suggest that the GPT-generated data aligns well with SiMT human preferences.

Table 5: Human evaluation on GPT-generated translations and manually revised translations.

| References | Average Score | Win-Tie-Lose | | | Distribution of Scores | | | |
|---|---|---|---|---|---|---|---|---|
| | | win ratio | lose ratio | tie ratio | 0 | 2 | 4 | 6 |
| *Chinese-English* | | | | | | | | |
| GPT-4/4o | 5.37 | 6.00% | 12.00% | 82.00% | 0.50% | 5.50% | 19.00% | 75.00% |
| Manually Revised | 5.57 | 12.00% | 6.00% | 82.00% | 0.00% | 1.00% | 19.50% | 79.50% |
| *German-English* | | | | | | | | |
| GPT-4/4o | 4.47 | 2.00% | 80.50% | 17.50% | 5.50% | 19.50% | 21.00% | 54.00% |
| Manually Revised | 5.01 | 17.50% | 80.50% | 2.00% | 3.00% | 10.00% | 20.50% | 66.50% |
| *English-Chinese* | | | | | | | | |
| GPT-4/4o | 4.61 | 1.00% | 94.00% | 5.00% | 3.50% | 10.50% | 38.00% | 48.00% |
| Manually Revised | 4.73 | 5.00% | 94.00% | 1.00% | 3.50% | 8.50% | 36.00% | 52.00% |

# B  PROOFS AND DERIVATIONS

## B.1  PROOF OF EQUIVALENCE BETWEEN OUTPUT LENGTH CONSTRAINT AND LATENCY OPTIMIZATION

To incorporate the goal of reducing latency into the optimization objective, we can directly include a latency evaluation metric. Specifically, we integrate Average Lagging (AL) (Ma et al., 2019), a

commonly used metric for measuring SiMT latency, into the optimization objective as follows:

$$\max_{\pi_\theta} \mathbb{E}_{x \sim D, y \sim \pi_\theta(y|x)}[r(x,y)] - \beta \mathbb{D}_{KL}[\pi_\theta(y \mid x)||\pi_{ref}(y \mid X)] - \mathbb{E}[AL] \tag{12}$$

Given the sample $(X, Y)$, when receiving source prefix x, the SiMT model needs to output the target prefix y. We assume that the SiMT model can accept a source sentence with a maximum length of $M$ and generate a target sentence with a maximum length of $N$. Therefore, the following relationship holds:

$$0 < |x| \leq |X| \leq M, \quad 0 < |y| \leq |Y| \leq N, \tag{13}$$

During simultaneous translation, the prefix pair $(x, y)$ evolves from $(x_0, \emptyset)$ to $(X, Y)$. For the SiMT process passing by $(x, y)$, the maximum possible latency occurs when the SiMT model waits for the full input of x before starting to generate y, then waits for the complete X before outputting the remaining part of Y. In this situation, AL can be computed as:

$$AL = \frac{1}{y}\sum_{t=1}^{|y|}(|x| - \frac{t-1}{\frac{|Y|}{|X|}}) = |x| - \frac{|X|}{2|Y|}(|y|-1) \leq (-\frac{1}{2N}|y| + \frac{1}{2N} + 1)|X| = -C_1|y| + C_2 \tag{14}$$

where $C_1 = \frac{|X|}{2N}, C_2 = \frac{|X|}{2N} + |X|$. Therefore, we can derive the following relationship:

$$\begin{aligned}
&\mathbb{E}_{x \sim D, y \sim \pi_\theta(y|x)}[r(x,y)] - \beta \mathbb{D}_{KL}[\pi_\theta(y \mid x)||\pi_{ref}(y \mid X)] - \mathbb{E}[AL] \\
&\leq \mathbb{E}_{x \sim D, y \sim \pi_\theta(y|x)}[r(x,y)] - \beta \mathbb{D}_{KL}[\pi_\theta(y \mid x)||\pi_{ref}(y \mid X)] + C_1\mathbb{E}[|y|] - C_2
\end{aligned} \tag{15}$$

Based on this upper bound, we can optimize the objective in Equation 12 by optimizing the following one:

$$\max_{\pi_\theta} \mathbb{E}_{x \sim D, y \sim \pi_\theta(y|x)}[r(x,y)] - \beta \mathbb{D}_{KL}[\pi_\theta(y \mid x)||\pi_{ref}(y \mid X)] + C_1\mathbb{E}[|y|] - C_2 \tag{16}$$

Since $C_1$ and $C_2$ are constants, this objective is equivalent to:

$$\max_{\pi_\theta} \mathbb{E}_{x \sim D, y \sim \pi_\theta(y|x)}[r(x,y)] - \beta \mathbb{D}_{KL}[\pi_\theta(y \mid x)||\pi_{ref}(y \mid X)] + \alpha \mathbb{E}[|y|] \tag{17}$$

where $\alpha$ is the added hyper-parameter.

## B.2 DERIVATION OF THE OPTIMAL SOLUTION FOR REWARD MAXIMIZATION CONSTRAINED BY KL DIVERGENCE AND LATENCY

Starting from Equation 6, we can conduct the derivation as follows:

$$\begin{aligned}
&\max_{\pi_\theta} \mathbb{E}_{x \sim D, y \sim \pi_\theta(y|x)}[r(x,y)] - \beta \mathbb{D}_{KL}[\pi_\theta(y \mid x)||\pi_{ref}(y \mid X)] + \alpha \mathbb{E}[|y|] \\
&= \max_{\pi_\theta} \mathbb{E}_{x \sim D}\mathbb{E}_{y \sim \pi_\theta(y|x)}[r(x,y) - \beta \log \frac{\pi_\theta(y \mid x)}{\pi_{ref}(y \mid X)} + \alpha|y|] \\
&= -\min_{\pi_\theta} \mathbb{E}_{x \sim D}\mathbb{E}_{y \sim \pi_\theta(y|x)}[\log \frac{\pi_\theta(y \mid x)}{\frac{1}{Z(x)}\pi_{ref}(y \mid X)\exp(\frac{1}{\beta}(r(x,y) + \alpha|y|))} - \log Z(x)] \\
&= -\min_{\pi_\theta} \mathbb{E}_{x \sim D}[\mathbb{D}_{KL}[\pi_{ref}||\frac{1}{Z(x)}\pi_{ref}(y \mid X)\exp(\frac{1}{\beta}(r(x,y) + \alpha|y|))] - \log Z(x)]
\end{aligned} \tag{18}$$

Where $Z(x)$ is the partition function, which is calculated as:

$$Z(x) = \sum_y \pi_{ref}(y \mid X)\exp(\frac{1}{\beta}(r(x,y) + \alpha|y|)) \tag{19}$$

Based on the property of KL divergence, we can gain the optimal solution for Equation 6 as:

$$\pi^* = \frac{1}{Z(x)}\pi_{ref}(y \mid X)\exp(\frac{1}{\beta}(r(x,y) + \alpha|y|)) \tag{20}$$

Thus, we can derive the expression of $r(x, y)$ concerning $\pi_\theta$ and $|y|$:

$$r(x,y) = \beta \log \frac{\pi_\theta(y \mid x)}{\pi_{ref}(y \mid X)} + \beta \log Z(x) - \alpha|y| \tag{21}$$

## C   DISTINCTIONS BETWEEN SIMULDPO AND R-DPO

Firstly, the objectives and methods of R-DPO are entirely opposite to SimulDPO. In R-DPO, Park et al. (2024) aim to prevent models from generating too long responses and use a regularization term of "$-\alpha|y|$" to achieve this. In contrast, for the SiMT task, audiences prefer translations with low latency, which requires the SiMT model to translate as much content as possible based on the already received source prefix. To achieve this, SimulDPO introduces "$+\alpha|y|$" as an additional constraint. It is important to note that the goal of SimulDPO is not to optimize for the length itself, but rather to optimize for latency preferences.

Secondly, as shown in Equation 9, we use $|y| = \sum_{t=1}^{|y|+1} c_t$ to make "$+\alpha|y|$" differentiable, allowing gradient signals to be directly propagated to the parameters through backpropagation. In contrast, Park et al. (2024) treats the "$-\alpha|y|$" as a margin without further processing.

## D   CONFIDENCE-BASED POLICY DURING INFERENCE

Algorithm 1 further illustrates the confidence-based policy adopted by SimulPL during inference. As shown in Algorithm 1, in the confidence-based policy, we set the confidence threshold to 0.5 as the basis for read/write decisions. To examine its impact, we compare the performance of SimulPL on the Zh→En task with different threshold values ($\gamma$). The results are presented in Table 6. When $\gamma$ is set to a small value ($\gamma$=0.1), the model is allowed to output tokens with low confidence, which results in a decline in translation quality, especially in low latency levels ($0 \leq$ LAAL $< 4$). When $\gamma$ is set to a higher value ($\gamma$=0.9), the model imposes stricter constraints on token quality, leading to unnecessary delays. We plan to further explore the impact of $\gamma$ in our future work.

---

**Algorithm 1:** Confidence-based Policy In Inference

---

**Input**   : Streaming source prefix x,$t = 1$,read length $n$, $y_0 \leftarrow \langle$BOS$\rangle$
**Output:** Target outputs $\mathbf{Y}$
1 **while** $y_{t-1} \neq \langle$EOS$\rangle$ **do**
2      estimate confidence $c_t$; **if** $c_t \geq 0.5$ **then**
3          generate $y_t$ with $\mathbf{x}, y_{\leq t-1}$;
4          $t \leftarrow t + 1$;
5      **else**
6          wait for next $n$ source words;
7          update x;
8      **end**
9 **end**

---

Table 6: Human evaluation on GPT-generated translations and manually revised translations.

| $n$ | $\gamma$=0.1 | | $\gamma$=0.3 | | $\gamma$=0.5 | | $\gamma$=0.7 | | $\gamma$=0.9 | |
|---|---|---|---|---|---|---|---|---|---|---|
| | **LAAL** | **SacreBLEU** | **LAAL** | **SacreBLEU** | **LAAL** | **SacreBLEU** | **LAAL** | **SacreBLEU** | **LAAL** | **SacreBLEU** |
| 3 | 2.03 | 20.76 | 1.99 | 21.51 | 1.73 | 22.51 | 1.99 | 22.36 | 2.04 | 22.51 |
| 5 | 3.21 | 24.25 | 3.19 | 24.89 | 2.90 | 24.94 | 3.11 | 25.24 | 3.29 | 24.97 |
| 7 | 4.36 | 26.48 | 4.36 | 26.58 | 4.04 | 26.76 | 4.28 | 26.94 | 4.45 | 26.78 |
| 10 | 6.27 | 28.17 | 6.25 | 28.10 | 5.87 | 27.97 | 6.18 | 28.45 | 6.36 | 27.81 |
| 15 | 9.42 | 29.86 | 9.40 | 29.80 | 9.08 | 29.94 | 9.22 | 30.13 | 9.46 | 30.20 |
| 20 | 12.37 | 30.17 | 12.35 | 30.20 | 12.01 | 30.05 | 12.15 | 30.52 | 12.46 | 30.47 |

## E   ADDITIONAL IMPLEMENTATION DETAILS

The hyper-parameters of the Transformer-based SiMT models used in our experiments are shown in Table 7. The hyper-parameters used by SimulPL during the MSFT and SimulDPO phases are listed in Table 8. In the training process, we share the LoRA in the MSFT and SimulDPO phases. We use the instruction-following format to guide the LLM in completing the SiMT task. Our used prompt

template is shown in Figure 10. For a fair comparison between Transformer-based and LLM-based SiMT models, we following Ma et al. (2020b) to apply greedy search during inference.

## F  NUMERICAL RESULTS

Tables 9, 10, and 11 respectively present the numerical results of different SiMT models on the Zh→En, De→En, and En→Zh SiMT tasks. In addition to LAAL, we also recorded other common latency metrics such as AL (Ma et al., 2019), DAL (Ma et al., 2020b), and AP (Cho, 2016).

## G  ADDITIONAL ANALYSES ON DE→EN SIMT TASK

In this section, we conduct our analysis experiments on De→En task. The results are respectively shown in Figure 11, 12, and 13. Through these experiments, we further validate our findings: Both SimulDPO and MSFT improve model performance, with a more pronounced effect at low latency levels; although the effect is less pronounced in the De→En task compared to the Zh→En task, SimulPL's performance is still influenced by $\alpha$; SimulPL also generalizes well to other preference optimization methods on the De→En task.

```
[INST] <<SYS>>
You are a professional translator.
<</SYS>>

translate the following text from <SOURCE_LANG> to <TARGET_LANG>:

<SOURCE_LANG>: <SOURCE_PREFIX>
<TARGET_LANG>: [/INST] <TARGET_PREFIX>
```

Figure 10: Our prompt template in the SimulPL framework and other LLM-based SiMT models.

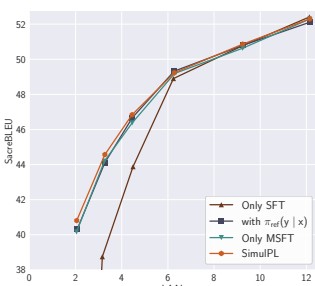

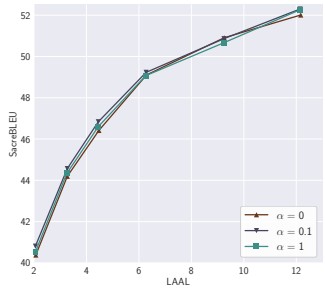

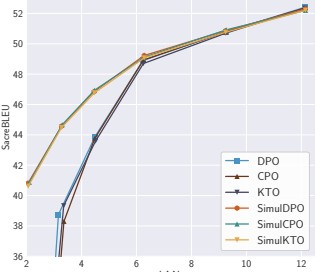

Figure 11:  Ablation Studies of SimulPL framework on De→En SiMT task.

Figure 12: The comparison of SimulDPO with difference $\alpha$ values on De→En SiMT task.

Figure 13:  The comparison between different preference optimization methods on De→En SiMT task.

Table 7: Hyper-parameters of Transformer-based SiMT models in our experiments.

| Transformer Hyper-parameter | |
|---|---|
| encoder layers | 6 |
| encoder attention heads | 8 |
| encoder embed dim | 512 |
| encoder ffn embed dim | 1024 |
| decoder layers | 6 |
| decoder attention heads | 8 |
| decoder embed dim | 512 |
| decoder ffn embed dim | 1024 |
| dropout | 0.1 |
| optimizer | adam |
| adam-$\beta$ | (0.9, 0.98) |
| clip-norm | 1e-7 |
| lr | 5e-4 |
| lr scheduler | inverse sqrt |
| warmup-updates | 4000 |
| warmup-init-lr | 1e-7 |
| weight decay | 0.0001 |
| label-smoothing | 0.1 |
| max tokens | 8192 |

Table 8: Hyper-parameters of SimulPL in our experiments.

| SimulPL Hyper-parameter | | |
|---|---|---|
| LoRA | lora_r | 64 |
| | lora_alpha | 16 |
| | lora_dropout | 0.1 |
| MSFT | batch_size | 64 |
| | micro_batch_size | 32 |
| | learning_rate | 2e-4 |
| | training steps | 1000 |
| SimulDPO | $\alpha$ | 0.1 |
| | $\beta$ | 0.1 |
| | batch_size | 64 |
| | micro_batch_size | 16 |
| | learning_rate | 2e-6 |
| | training steps | 400 |

Table 9: Numerical results on Zh→En SiMT task.

| **Chinese→English** | | | | | | |
|---|---|---|---|---|---|---|
| wait-$k$ | | | | | | |
| $k$ | BLEU | COMET | LAAL | AL | AP | DAL |
| 1 | 9.70 | 56.42 | 2.02 | 1.04 | 0.48 | 2.53 |
| 3 | 17.88 | 68.80 | 4.22 | 2.92 | 0.69 | 4.93 |
| 5 | 21.93 | 73.90 | 6.46 | 4.70 | 0.88 | 7.27 |
| 7 | 24.79 | 75.96 | 8.48 | 6.60 | 1.00 | 9.34 |
| 9 | 25.73 | 76.48 | 10.26 | 8.54 | 1.05 | 11.11 |
| SM$^2$ | | | | | | |
| $\gamma$ | BLEU | COMET | LAAL | AL | AP | DAL |
| 0.3 | 21.96 | 75.43 | 3.99 | 2.41 | 0.75 | 7.29 |
| 0.4 | 25.87 | 77.82 | 5.44 | 3.65 | 0.86 | 9.51 |
| 0.5 | 27.94 | 78.75 | 6.80 | 5.06 | 0.93 | 11.73 |
| 0.55 | 28.60 | 79.06 | 7.55 | 5.95 | 0.97 | 13.07 |
| 0.6 | 28.83 | 79.33 | 8.64 | 7.17 | 1.01 | 14.99 |
| 0.65 | 29.04 | 79.44 | 10.23 | 8.98 | 1.07 | 17.33 |
| 0.7 | 29.10 | 79.55 | 12.41 | 11.45 | 1.13 | 20.69 |
| LLM-PFX-SFT | | | | | | |
| $n$ | BLEU | COMET | LAAL | AL | AP | DAL |
| 3 | 10.07 | 71.98 | 3.96 | -13.82 | 1.24 | 7.81 |
| 5 | 14.02 | 76.51 | 4.30 | -14.45 | 1.12 | 8.12 |
| 7 | 22.01 | 78.13 | 4.59 | -4.10 | 0.85 | 8.99 |
| 10 | 26.24 | 79.39 | 6.11 | 0.57 | 0.84 | 11.39 |
| 15 | 29.54 | 80.45 | 9.21 | 5.99 | 0.88 | 15.20 |
| 20 | 30.08 | 80.83 | 12.06 | 9.53 | 0.93 | 18.51 |
| LLM-PFX-SFT + DPO | | | | | | |
| $n$ | BLEU | COMET | LAAL | AL | AP | DAL |
| 3 | 12.08 | 69.39 | 3.39 | -10.22 | 1.03 | 6.54 |
| 5 | 17.49 | 75.95 | 3.56 | -9.87 | 0.93 | 7.16 |
| 7 | 24.44 | 78.44 | 4.27 | -2.35 | 0.79 | 8.48 |
| 10 | 27.35 | 79.84 | 5.96 | 2.59 | 0.80 | 11.08 |
| 15 | 29.12 | 81.06 | 9.14 | 6.10 | 0.87 | 15.12 |
| 20 | 29.82 | 81.44 | 12.04 | 9.45 | 0.93 | 18.49 |
| SimulPL | | | | | | |
| $n$ | BLEU | COMET | LAAL | AL | AP | DAL |
| 3 | 22.51 | 75.20 | 1.73 | -3.25 | 0.68 | 4.50 |
| 5 | 24.94 | 77.44 | 2.90 | -1.31 | 0.70 | 6.43 |
| 7 | 26.76 | 78.79 | 4.04 | 0.42 | 0.73 | 8.32 |
| 10 | 27.97 | 79.78 | 5.87 | 2.42 | 0.79 | 11.05 |
| 15 | 29.94 | 80.85 | 9.08 | 6.21 | 0.86 | 15.11 |
| 20 | 30.05 | 81.14 | 12.01 | 9.65 | 0.91 | 18.49 |

Table 10: Numerical results on De→En SiMT task.

| **German→English** | | | | | | |
|---|---|---|---|---|---|---|
| | | | wait-$k$ | | | |
| $k$ | BLEU | COMET | LAAL | AL | AP | DAL |
| 1 | 12.77 | 56.08 | 0.68 | 0.06 | 0.46 | 1.54 |
| 3 | 31.11 | 69.52 | 2.98 | 1.40 | 0.87 | 4.11 |
| 5 | 39.24 | 76.62 | 4.98 | 3.28 | 1.04 | 6.14 |
| 7 | 41.42 | 79.41 | 7.21 | 5.05 | 1.23 | 8.25 |
| 9 | 42.59 | 80.36 | 9.02 | 6.95 | 1.32 | 9.99 |
| | | | $SM^2$ | | | |
| $\gamma$ | BLEU | COMET | LAAL | AL | AP | DAL |
| 0.3 | 38.76 | 80.25 | 4.06 | 2.43 | 1.05 | 7.62 |
| 0.4 | 43.63 | 82.27 | 5.21 | 3.30 | 1.18 | 9.27 |
| 0.5 | 44.99 | 83.22 | 6.49 | 4.52 | 1.29 | 10.94 |
| 0.55 | 45.79 | 83.63 | 7.23 | 5.25 | 1.35 | 11.95 |
| 0.6 | 46.05 | 83.77 | 8.06 | 6.21 | 1.40 | 12.97 |
| 0.65 | 46.47 | 83.99 | 9.05 | 7.40 | 1.45 | 14.23 |
| 0.7 | 46.82 | 84.18 | 10.27 | 8.85 | 1.52 | 15.76 |
| | | | LLM-PFX-SFT | | | |
| $n$ | BLEU | COMET | LAAL | AL | AP | DAL |
| 3 | 19.05 | 79.46 | 2.58 | -11.35 | 1.32 | 4.98 |
| 5 | 40.16 | 82.95 | 3.37 | -0.16 | 0.85 | 6.31 |
| 7 | 44.04 | 84.25 | 4.48 | 1.56 | 0.86 | 7.98 |
| 10 | 48.58 | 85.47 | 6.24 | 3.98 | 0.90 | 10.34 |
| 15 | 50.68 | 86.30 | 9.22 | 7.36 | 0.97 | 13.65 |
| 20 | 52.13 | 86.66 | 12.11 | 10.56 | 1.02 | 16.13 |
| | | | LLM-PFX-SFT+DPO | | | |
| $n$ | BLEU | COMET | LAAL | AL | AP | DAL |
| 3 | 19.82 | 79.52 | 2.46 | -9.20 | 1.25 | 4.79 |
| 5 | 38.75 | 82.86 | 3.16 | -0.43 | 0.85 | 6.28 |
| 7 | 43.88 | 84.16 | 4.49 | 1.82 | 0.85 | 8.03 |
| 10 | 48.90 | 85.41 | 6.24 | 4.16 | 0.88 | 10.34 |
| 15 | 50.78 | 86.18 | 9.25 | 7.57 | 0.95 | 13.67 |
| 20 | 52.41 | 86.27 | 12.13 | 10.77 | 1.00 | 16.14 |
| | | | SimulPL | | | |
| $n$ | BLEU | LAAL | AL | AP | DAL | COMET |
| 3 | 40.81 | 2.05 | -1.43 | 0.74 | 4.22 | 81.15 |
| 5 | 44.57 | 3.27 | 0.42 | 0.78 | 6.14 | 83.53 |
| 7 | 46.84 | 4.45 | 2.00 | 0.82 | 7.90 | 84.44 |
| 10 | 49.23 | 6.28 | 4.12 | 0.88 | 10.32 | 85.41 |
| 15 | 50.86 | 9.24 | 7.49 | 0.95 | 13.65 | 86.27 |
| 20 | 52.30 | 12.15 | 10.69 | 1.00 | 16.13 | 86.53 |

Table 11: Numerical results on En→Zh SiMT task.

| | BLEU | COMET | LAAL | AL | AP | DAL |
|---|---|---|---|---|---|---|
| **English→Chinese** | | | | | | |
| wait-$k$ | | | | | | |
| $k$ | BLEU | COMET | LAAL | AL | AP | DAL |
| 1 | 6.37 | 57.00 | 0.59 | 0.36 | 0.33 | 1.17 |
| 3 | 26.09 | 70.28 | 2.77 | 1.83 | 0.75 | 3.54 |
| 5 | 35.52 | 76.39 | 4.80 | 3.59 | 0.96 | 5.52 |
| 7 | 38.46 | 78.91 | 6.73 | 5.45 | 1.08 | 7.35 |
| 9 | 40.76 | 80.00 | 8.37 | 7.32 | 1.12 | 8.94 |
| $SM^2$ | | | | | | |
| $\gamma$ | BLEU | COMET | LAAL | AL | AP | DAL |
| 0.3 | 39.46 | 79.83 | 4.52 | 3.38 | 0.91 | 6.81 |
| 0.4 | 41.96 | 81.01 | 5.31 | 4.19 | 0.97 | 7.77 |
| 0.5 | 42.81 | 81.64 | 6.13 | 5.13 | 1.01 | 8.82 |
| 0.6 | 43.27 | 81.97 | 7.38 | 6.53 | 1.08 | 10.52 |
| 0.65 | 43.63 | 82.14 | 8.30 | 7.61 | 1.12 | 11.92 |
| 0.7 | 43.90 | 82.22 | 9.62 | 9.08 | 1.16 | 13.60 |
| LLM-PFX-SFT | | | | | | |
| $n$ | BLEU | COMET | LAAL | AL | AP | DAL |
| 3 | 7.58 | 69.86 | 4.43 | -57.94 | 4.02 | 7.43 |
| 5 | 30.88 | 79.61 | 4.40 | -1.91 | 1.14 | 7.19 |
| 7 | 40.31 | 81.87 | 5.46 | 3.92 | 0.92 | 8.45 |
| 10 | 43.62 | 82.96 | 7.29 | 6.47 | 0.91 | 10.55 |
| 15 | 44.47 | 83.54 | 10.00 | 9.48 | 0.96 | 13.26 |
| 20 | 44.47 | 83.58 | 12.23 | 11.84 | 1.00 | 15.17 |
| LLM-PFX-SFT+DPO | | | | | | |
| $n$ | BLEU | COMET | LAAL | AL | AP | DAL |
| 3 | 7.85 | 69.28 | 4.42 | -54.99 | 3.87 | 7.41 |
| 5 | 31.39 | 79.81 | 4.26 | -0.77 | 1.04 | 6.92 |
| 7 | 39.95 | 81.41 | 5.47 | 3.88 | 0.92 | 8.48 |
| 10 | 43.21 | 82.33 | 7.26 | 6.45 | 0.91 | 10.52 |
| 15 | 44.08 | 83.05 | 10.01 | 9.49 | 0.96 | 13.28 |
| 20 | 44.12 | 83.10 | 12.23 | 11.83 | 0.99 | 15.17 |
| SimulPL | | | | | | |
| $n$ | BLEU | COMET | LAAL | AL | AP | DAL |
| 3 | 37.26 | 79.53 | 2.64 | 1.24 | 0.73 | 4.87 |
| 5 | 40.72 | 81.51 | 3.97 | 2.86 | 0.79 | 6.58 |
| 7 | 42.30 | 81.77 | 5.32 | 4.28 | 0.87 | 8.33 |
| 10 | 43.86 | 82.81 | 7.18 | 6.37 | 0.90 | 10.46 |
| 15 | 44.83 | 83.58 | 9.89 | 9.33 | 0.96 | 13.21 |
| 20 | 45.09 | 83.68 | 12.18 | 11.75 | 1.00 | 15.16 |

