# OpenReview forum: "SimulPL: Aligning Human Preferences in Simultaneous Machine Translation"
_ICLR.cc/2025/Conference — ICLR 2025 Poster_

### Official Review · Reviewer_1TVX · 2024-10-28

**Soundness:** 4
**Presentation:** 4
**Contribution:** 3
**Rating:** 8
**Confidence:** 4

**Summary:**

This paper introduces a new approach to simultaneous interpretation with low latency.

**Strengths:**

The experimental results look impressive.

I think there will also be interest in the experimental materials.  Your arguments about offline machine
translation (OMT)  are convincing.  That said, I didn't see any discussion of sharing those materials.

I like the idea of offering the user choices over preferences, but do we need to restrict ourselves to a single choice?  Could you imagine a speech-to-text scenario where you could offer multiple text windows so the user can look at one window when latency is a priority and another window when BLEU is a priority?

**Weaknesses:**

I have never been that happy with the standard discussion of latency and BLEU scores in this literature.  I have quite a bit of experience with professional interpreters as well as automatic solutions.  According to a number of standard metrics, the automatic solutions are better than professional interpreters, but I know that I am more engaged in the meeting when I have access to an interpreter.  They tell me what I need to know when I need to know it.  In addition, sometimes the speaker uses a metaphor that requires more explanation.    The professionals understand when then need to do more than just translate what the speaker said.

I never liked average latency.  I can't fault you for doing what others have done, but sometimes latency matters and sometimes it doesn't.  The professionals understand this.  They speed up when necessary and slow down when latency is less of a priority.  Moreover, I probably care more about worst case latency or RMS latency than average latency.

This literature would benefit by including more input from professional interpreters that have a lot more experience in this area than we have.

I would really like to see some data from schools that teach professional interpreters. We should be able to record what students say in the booth when they learn how to do this task.

**Questions:**

Do you have any plans to share your benchmark?

Do you have a github that will make it easy for others to replicate your work?

---

> ### Author Response · Authors · 2024-11-20
> **Thanks for Reviewer 1TVX's valuable comments! Here is the response to Reviewer 1TVX**
>
> Thank you for your positive feedback on our work, which is truly encouraging! We also appreciate your valuable suggestions. The following are our detailed responses.
>
>
> **Q1:** Regarding the sharing of experimental materials.
>
> **A1:** Thank you for your consideration of our work. We are organizing our code, datasets, and other experimental materials. We plan to share these materials with the community after our paper is published.
>
> **Q2:** Regarding multiple translation options.
>
> **A2:** Thank you for your interesting suggestion. Offering users the choice between latency-prioritized and quality-prioritized translations would indeed better accommodate the diverse needs of different users. In practical applications, we can run multiple SimulPL models in parallel with different hyper-parameter values of $n$, which would allow us to generate translations under varying latency conditions and fulfill the scenario you proposed. We will explore this point further in our future work.
>
> **Q3:** Discussion on latency evaluation.
>
> **A3:** Your comments on latency evaluation are insightful. Existing evaluation metrics based on average latency indeed fails to fully reflect the worst-case latency, and the RMS latency also provides useful reference. We appreciate your understanding of our use of common metrics, as it allows for easier and fairer comparison with previous work. We will explore more appropriate evaluation metrics for latency in our future work.
>
> **Q4:** Collaboration with professional interpreters for further research.
>
> **A4:** Thank you for your suggestion. In our study, both the manual evaluation of the training data and the manual revision of the test set were performed by professional simultaneous interpreters, which greatly helped us to improve the dataset's quality. As you suggested, we will consider collecting data from institutions specializing in interpreting education in our future work.
>
> **Q5:** The sharing of the benchmark and GitHub repo.
>
> **A5:** Yes, we plan to release our benchmark. We are organizing our code and data and will provide the GitHub link in the preprint of our paper.
>
> **Thank you again for your recognition of our work and your insightful suggestions! We truly appreciate your deep understanding and unique perspective on the SiMT task. We are committed to incorporating your suggestions into our future work.**

---

> > ### Comment · Reviewer_1TVX · 2024-11-26
> > **nothing more to add**
> >
> > I'm happy with the authors' responses to my review.  As for the advice on how to review, I clicked that I was a first time reviewer for this venue (which may be correct), though I have reviewed for many other venues.  The authors seem to have taken my hints seriously.  I cannot fault them for following standard practices, though I did want to call out that we should rethink some of those standard practices.  The authors seem aware of these concerns.  I am confident they will do what they can to address these concerns in future work.  That is about the most that one can expect.

---

> > > ### Author Response · Authors · 2024-11-27
> > > **Thank you for your further feedback!**
> > >
> > > We appreciate your additional feedback. Once again, we sincerely thank you for your recognition of our work and the trust you have placed in us. This truly means a lot to us.

---

### Official Review · Reviewer_qWwc · 2024-10-31

**Soundness:** 2
**Presentation:** 3
**Contribution:** 3
**Rating:** 6
**Confidence:** 4

**Summary:**

This paper studies the problem of Simultaneous Machine Translation (SiMT). To achieve preference alignment for SiMT models, this paper categorizes human preferences in SiMT scenarios and focus on five aspects: translation quality preference, monotonicity preference, key point preference, simplicity preference, and latency preference. Based on this catogorization, this pape rconstructs human preference prompts with GPT to generate preference data for SiMT. After constructing the data, this paper designs a multi-task supervised fine-tuning stage and a simultaneous direct preference optimization stage to achieve preference alignment for the SiMT model. Experiments on text-to-text SiMT tasks show the effectiveness of the proposed approach.

**Strengths:**

1. The proposed approach shows performance improvement on the machine translation datasets.
2. The proposed five aspects for SiMT preference alignment could be useful.
3. The presentation is clear and easy to understand.

**Weaknesses:**

1. Heavy reliance on the quality of the GPT-generated preference data, and few effort is done to verify the quality/correctness/trustworthiness of the generated data.
2. Overall limited novelty for the whole framework, the idea of length penalty term |y| in the proposed SimulDPO loss function come from [1], and nothing much novel is added here.
3. Scope is very limited to SiMT only, and this method does not seem applicable to more general LLM preference alignment cases.
4. Some design is arbitrary. For example, in Confidence-Based Policy During Inference, why set the confidence c_t as exactly 0.5? What if your confidence estimation model is biased, e.g., $\mathbb{E}(c_t)\neq0.5$?
5. More experiments and analysis should be done to conduct a comprehensive evaluation. In the current form, experiment analysis are limited.


References:
[1] Park, Ryan, et al. "Disentangling length from quality in direct preference optimization." arXiv preprint arXiv:2403.19159 (2024).

**Questions:**

Why most experiments are done between translation of Zh and En? Should possibly incorporate more languages to demonstrate the effectiveness of the approach.

---

> ### Author Response · Authors · 2024-11-20
> **Thanks for Reviewer qWwc's valuable comments! Here is the response to Reviewer qWwc (Part I)**
>
> Thank you for your valuable comments. We fully understand your concerns and questions. The following are our detailed responses.
>
> **Q1**: Validation of our constructed data quality.
>
> **A1**: *Firstly*, we conduct both human evaluation and automatic multi-aspect evaluation for GPT-generated training set. In the human evaluation, we employ professional simultaneous interpreters to assess the translations generated by GPT-4/4o. The results in Figure 2 indicate that the GPT-generated translations aligns better with human preferences compared to the original references. In the  multi-aspect evaluation, we assess the translations with Ref-free COMET (Table 1) as well as our defined NIR, DD, and SLR metrics (Table 3 in Appendix A.2). These results demonstrate that the GPT-generated translations outperform the original references in terms of Translation Quality Preference, Monotonicity Preference, Key Point Preference, and Simplicity Preference. We will emphasize these experimental findings in the revised version of our paper.
>
> *Secondly*, when constructing the test set, we ask professional simultaneous interpreters to manually revise the GPT-generated drafts to produce the final annotations, which better align with human preferences. This is mentioned in lines 221-222 on Page 5 of our paper.
>
> *Thirdly*, to further validate the quality of our data and address your concerns, we add a comparative analysis of GPT-generated translations and those manually revised by interpreters on the test set. Please refer to our **General Response: Further Evaluation of GPT-generated Data**.
>
> We will include our new analysis experiments in the revised version of our paper, hoping that these could alleviate your concerns.
>
> **Q2**: Our novel contributions & Distinctions from related work.
>
> **A2**: **Our Novel Contributions:** Our contribution is not limited to SimulDPO. Instead, we propose a comprehensive preference learning framework (SimulPL) tailored to the SiMT task, which includes:
>
> 1. **Categorization of SiMT human preferences**: We categorize human preferences in SiMT scenarios into five aspects: translation quality, monotonicity, key points, simplicity, and latency. This bridges the gap in the study of human preferences in SiMT task (Section 4.1).
> 2. **Efficient preference dataset construction with human preference prompts**: By leveraging our categorized human preferences, we can create human preference prompts, which enable the efficient construction of dataset that aligns with human preferences using LLMs like GPT (Section 4.2).
> 3. **MSFT and SimulDPO in the training process**: Existing preference optimization methods neither consider latency preferences in SiMT nor optimize the read/write policy. In contrast, SimulPL addresses these issues. In MSFT phase (Section 4.3), SimulPL jointly learns the translation ability and read/write policy for initial preference alignment. In SimulDPO phase (Section 4.4), SimulPL can not only incorporate latency preferences into the training objectives but also integrate the read/write policy into the optimization process.
>
> Therefore, we believe your concerns regarding the novelty of our work are unnecessary. We have outlined these contributions in our Introduction and Conclusion, and we will further emphasize our contributions in the revised version of the paper.
>
> **Distinctions from Related Work:** *Firstly*, the objectives and methods of Park etal.[1] are entirely opposite to our proposed SimulPL. In Park etal. [1], the authors aim to prevent models from generating too long responses and use a regularization term of "$-\alpha |y|$" to achieve this. In contrast, for the SiMT task, audiences prefer translations with low latency, which requires the SiMT model to translate as much content as possible based on the already received source prefix. To achieve this, we choose "$+\alpha |y|$" as an additional constraint (Equation 6 in our paper). **It is important to note that our goal is not to optimize for the length itself, but rather to optimize for latency preferences. We choose "$+\alpha |y|$" because it is equivalent to optimizing for latency preferences, as proven in Appendix B.1.**
>
> *Secondly*, we use $|\mathrm{y}| = \sum_{t=1}^{|\mathrm{y}|+1} c_t$ to make $|\mathrm{y}|$ differentiable, allowing gradient signals to be directly propagated to the parameters through backpropagation, as shown in line 290 and Equation 8 of our paper. In contrast, Park etal. [1] treats the "$-\alpha |y|$" as a margin without further processing.
>
> Thank you for the article you recommended. It is a valuable and interesting work. **We will cite the article you recommended and clarify these distinctions in the revised version of our paper.**
>
> **Reference**
>
>  [1] Park, Ryan, et al. "Disentangling length from quality in direct preference optimization." arXiv preprint arXiv:2403.19159 (2024).

---

> ### Author Response · Authors · 2024-11-20
> **Thanks for Reviewer qWwc's valuable comments! Here is the response to Reviewer qWwc (Part II)**
>
> **Q3:** Generalizability of SimulPL.
>
> **A3**: *Firstly*, our work focuses on aligning human preferences in SiMT task, rather than issue of general preference alignment in LLMs. We propose SimulPL to address the limitations of existing general preference learning methods in the SiMT scenarios. **We believe that solving preference alignment for specific tasks is equally important.**
>
> *Secondly*, SimulPL still maintains distinct generalizability. For instance, the SimulPL framework can be extended to simultaneous inference tasks (Chen et al. [1]), which represents a promising direction for reducing latency of LLM responses. We will provide further clarification on this point further in the revised version of our paper. We will also explore the preference alignment of simultanous inference tasks in our future work.
>
> **Reference**
>
> 【1】Chen, Chuangtao, et al. "LiveMind: Low-latency Large Language Models with Simultaneous Inference." *arXiv preprint arXiv:2406.14319* (2024).
>
> **Q4:** Threshold of confidence in the inference process.
>
> **A4:** Thank you for your suggestion regarding the threshold $\gamma$ for confidence $c_t$. We have conducted additional experiments analyzing the impact of different threshold $\gamma$. Specifically, we assess the performenace of SimulPL on  $\gamma=0.1,0.3,0.5,0.7,0.9$ with different $n$ during inference. Due to time constraints, we have only performed experiments on the Zh$\rightarrow$En task. The results are as follows:
>
> |      | $\gamma$=0.1 |           | $\gamma$=0.3 |           | $\gamma$=0.5 |           | $\gamma$=0.7 |           | $\gamma$=0.9 |           |
> | ---- | ------------ | --------- | ------------ | --------- | ------------ | --------- | ------------ | --------- | ------------ | --------- |
> | $n$  | LAAL         | SacreBLEU | LAAL         | SacreBLEU | LAAL         | SacreBLEU | LAAL         | SacreBLEU | LAAL         | SacreBLEU |
> | 3    | 2.03         | 20.76     | 1.99         | 21.51     | 1.73         | 22.51     | 1.99         | 22.36     | 2.04         | 22.51     |
> | 5    | 3.21         | 24.25     | 3.19         | 24.89     | 2.90         | 24.94     | 3.11         | 25.24     | 3.29         | 24.97     |
> | 7    | 4.36         | 26.48     | 4.36         | 26.58     | 4.04         | 26.76     | 4.28         | 26.94     | 4.45         | 26.78     |
> | 10   | 6.27         | 28.17     | 6.25         | 28.10     | 5.87         | 27.97     | 6.18         | 28.45     | 6.36         | 27.81     |
> | 15   | 9.42         | 29.86     | 9.40         | 29.80     | 9.08         | 29.94     | 9.22         | 30.13     | 9.46         | 30.20     |
> | 20   | 12.37        | 30.17     | 12.35        | 30.20     | 12.01        | 30.05     | 12.15        | 30.52     | 12.46        | 30.47     |
>
> When $\gamma$ is set to a small value ($\gamma=0.1$), the model is allowed to output tokens with low confidence, which results in a decline in translation quality, especially in low latency levels (LAAL < 4). When $\gamma$ is set to a higher value ($\gamma=0.9$), the model imposes stricter constraints on token quality, leading to unnecessary delays. We plan to further explore the impact of $\gamma$ in our future work.

---

> ### Author Response · Authors · 2024-11-20
> **Thanks for Reviewer qWwc's valuable comments! Here is the response to Reviewer qWwc (Part III)**
>
> **Q5:** Additional analysis experiments on more language pairs.
>
> **A5:** *First*, we conduct experiments on translation quality (Section 5.2) and preference evaluation (Section 5.3) with three language pairs: Zh$\rightarrow$En, En$\rightarrow$Zh, and De$\rightarrow$En. Results on SacreBLEU and COMET show that SimulPL can achieve better translation quality. Both human evaluation and automatic multi-aspect evaluation indicate that SimulPL aligns better with SiMT human preferences. We believe these experiments can sufficiently validate the effectiveness of SimulPL.
>
> *Second*, as for the analysis experiments, such as ablation studies (Section 5.4), impact of $\alpha$ (Section 5.5), and generalization to other preference optimiazation methods (Section 5.6), we believe our experimental findings can apply to other language pairs as SimulPL is language-agnostic.
>
> *Third*, we further validate these experiments on De$\rightarrow$En task. Specifically, the results of ablations studies are as follows:
>
> |      | SimulPL |           | Only MSFT |           | with  $\pi_\mathrm{ref}$ |           | SFT   |           |
> | ---- | ------- | --------- | --------- | --------- | ------------------------ | --------- | ----- | --------- |
> | $n$  | LAAL    | SacreBLEU | LAAL      | SacreBLEU | LAAL                     | SacreBLEU | LAAL  | SacreBLEU |
> | 3    | 2.05    | 40.81     | 2.04      | 40.16     | 2.07                     | 40.31     | 2.46  | 19.82     |
> | 5    | 3.27    | 44.57     | 3.25      | 44.19     | 3.27                     | 44.11     | 3.16  | 38.75     |
> | 7    | 4.45    | 46.84     | 4.45      | 46.37     | 4.46                     | 46.67     | 4.49  | 43.88     |
> | 10   | 6.28    | 49.23     | 6.29      | 49.19     | 6.27                     | 49.32     | 6.24  | 48.90     |
> | 15   | 9.24    | 50.86     | 9.24      | 50.63     | 9.24                     | 50.79     | 9.25  | 50.78     |
> | 20   | 12.15   | 52.30     | 12.14     | 52.29     | 12.15                    | 52.11     | 12.13 | 52.41     |
>
> The results analyzing the impact of $\alpha$ are as follows:
>
> |      | $\alpha=0$ |           | $\alpha=0.1$ |           | $\alpha=1$ |           |
> | ---- | ---------- | --------- | ------------ | --------- | ---------- | --------- |
> | $n$  | LAAL       | SacreBLEU | LAAL         | SacreBLEU | LAAL       | SacreBLEU |
> | 3    | 2.07       | 40.38     | 2.05         | 40.81     | 2.05       | 40.53     |
> | 5    | 3.27       | 44.20     | 3.27         | 44.57     | 3.25       | 44.34     |
> | 7    | 4.47       | 46.42     | 4.45         | 46.84     | 4.45       | 46.60     |
> | 10   | 6.29       | 49.11     | 6.28         | 49.23     | 6.26       | 49.07     |
> | 15   | 9.25       | 50.91     | 9.24         | 50.86     | 9.23       | 50.66     |
> | 20   | 12.17      | 52.01     | 12.15        | 52.30     | 12.14      | 52.26     |
>
> The results demonstrating SimulPL's generalization to other preference optimization methods are as follows:
>
> |      | CPO   |           | SimulCPO |           | KTO   |           | SimulKTO |           | DPO   |           | SimulDPO |           |
> | ---- | ----- | --------- | -------- | --------- | ----- | --------- | -------- | --------- | ----- | --------- | -------- | --------- |
> | $n$  | LAAL  | SacreBLEU | LAAL     | SacreBLEU | LAAL  | SacreBLEU | LAAL     | SacreBLEU | LAAL  | SacreBLEU | LAAL     | SacreBLEU |
> | 3    | 2.53  | 20.38     | 2.10     | 40.87     | 2.44  | 21.51     | 2.06     | 40.62     | 2.46  | 19.82     | 2.05     | 40.81     |
> | 5    | 3.36  | 38.30     | 3.30     | 44.67     | 3.35  | 39.33     | 3.27     | 44.48     | 3.16  | 38.75     | 3.27     | 44.57     |
> | 7    | 4.48  | 43.76     | 4.47     | 46.94     | 4.51  | 43.59     | 4.47     | 46.77     | 4.49  | 43.88     | 4.45     | 46.84     |
> | 10   | 6.25  | 48.93     | 6.29     | 49.13     | 6.25  | 48.70     | 6.28     | 49.07     | 6.24  | 48.90     | 6.28     | 49.23     |
> | 15   | 9.24  | 50.78     | 9.27     | 50.92     | 9.25  | 50.70     | 9.24     | 50.76     | 9.25  | 50.78     | 9.24     | 50.86     |
> | 20   | 12.11 | 52.35     | 12.15    | 52.20     | 12.13 | 52.35     | 12.15    | 52.21     | 12.13 | 52.41     | 12.15    | 52.30     |
>
> Through these experiments, we further validate our findings presented in the paper: (1) Both SimulDPO and MSFT improve model performance, with a more pronounced effect at low latency levels. (2) Although the effect is less pronounced in the De$\rightarrow$En task compared to the Zh$\rightarrow$En task, SimulPL's performance is still influenced by $\alpha$. (3) SimulPL also generalizes well to other preference alignment methods on the De$\rightarrow$En task.
>
> These results confirm the reliability of our analyses. We will include these experimental results in our revised paper and further supplement the experiments on the En$\rightarrow$Zh task.

---

> ### Author Response · Authors · 2024-11-20
> **Thanks for Reviewer qWwc's valuable comments! Here is the response to Reviewer qWwc (Part IV)**
>
> **Finally, thank you once again for your insightful suggestions! We also appreciate the paper you recommended in Q2, which we truly believe is a valuable contribution. We sincerely hope that our responses have addressed your concerns. If so, we would be grateful if you could consider recognizing our efforts and increasing your score. Should you have any further questions, please do not hesitate to reach out. We look forward to further discussion with you!**

---

> > ### Comment · Reviewer_qWwc · 2024-11-20
> >
> > Thank you for providing a comprehensive response! I think most of my concerns are addressed.

---

> ### Author Response · Authors · 2024-11-25
> **Thanks for Feedback!**
>
> We sincerely appreciate your recognition of our work and the increase in our score. It is an honor for us to address your concerns, and we promise to incorporate your invaluable insights into the final revised version of our paper.

---

### Official Review · Reviewer_TojP · 2024-11-04

**Soundness:** 3
**Presentation:** 2
**Contribution:** 2
**Rating:** 6
**Confidence:** 4

**Summary:**

The paper introduces Simultaneous Preference Learning (SimulPL) for the task of Simultaneous Machine Translation. The approach encompasses not only data construction but also model training and inference strategies. However, the novelty of the proposed method appears to be insufficient.

**Strengths:**

The paper presents a comprehensive approach to Simultaneous Machine Translation, addressing data construction, model training, and inference.

**Weaknesses:**

1. The paper lacks a clear emphasis on the innovative aspects of the proposed method. Highlighting specific novel contributions would strengthen the paper.

2. It remains unclear how the quality of the automatic labeling by GPT-4 is ensured, and the consistency rate with human labeling is not provided.

3. The distinction between the uppercase X and lowercase x in equations 6 and 7 needs clarification.

4. According to Figure 6, the improvement of SimulPL over only MSFT is not substantial, raising questions about the effectiveness of the proposed method.

**Questions:**

please see weaknesses

---

> ### Author Response · Authors · 2024-11-20
> **Thanks for Reviewer TojP's valuable comments! Here is the response to Reviewer TojP (Part I)**
>
> Thank you for your valuable comments! The following are our detailed responses.
>
> **Q1:** Highlighting our contributions.
>
> **A1:** Our contribution is that we propose SimulPL, a comprehensive preference learning framework tailored to the SiMT task, which includes:
>
> 1. **Categorization of SiMT human preferences**: We categorize human preferences in SiMT scenarios into five aspects: translation quality, monotonicity, key points, simplicity, and latency. This bridges the gap in the study of human preferences in SiMT task (Section 4.1).
> 2. **Efficient preference dataset construction with human preference prompts**: By leveraging our categorized human preferences, we can create human preference prompts, which enable the efficient construction of dataset that aligns with human preferences using LLMs like GPT (Section 4.2).
> 3. **MSFT and SimulDPO in the training process**: Existing preference optimization methods neither consider latency preferences in SiMT nor optimize the read/write policy. In contrast, SimulPL addresses these issues. In MSFT phase (Section 4.3), SimulPL jointly learns the translation ability and read/write policy for initial preference alignment. In SimulDPO phase (Section 4.4), SimulPL can not only incorporate latency preferences into the training objectives but also integrate the read/write policy into the optimization process.
>
> We outline our contributions in the Introduction and Conclusion. We will further emphasize these points in the revised version of our paper.
>
> **Q2**: Validation of our constructed data quality.
>
> **A2:** *Firstly*, we conduct both human evaluation and automatic multi-aspect evaluation for GPT-generated training set. In the human evaluation, we employ professional simultaneous interpreters to assess the translations generated by GPT-4/4o. The results in Figure 2 indicate that the GPT-generated translations aligns better with human preferences compared to the original references. In the  multi-aspect evaluation, we assess the translations with Ref-free COMET (Table 1) as well as our defined NIR, DD, and SLR metrics (Table 3 in Appendix A.2). These results demonstrate that the GPT-generated translations outperform the original references in terms of Translation Quality Preference, Monotonicity Preference, Key Point Preference, and Simplicity Preference. We will emphasize these experimental findings in the revised version of our paper.
>
> *Secondly*, when constructing the test set, we ask professional simultaneous interpreters to manually revise the GPT-generated drafts to produce the final annotations, which better align with human preferences. This is mentioned in lines 221-222 on Page 5 of our paper.
>
> *Thirdly*, to further validate the quality of our data and address your concerns, we add a comparative analysis of GPT-generated translations and those manually revised by interpreters on the test set. Please refer to our **General Response: Further Evaluation of GPT-generated Data**.
>
> We will include our new analysis experiments in the revised version of our paper, hoping that these could alleviate your concerns.
>
> **Q3**: Explanation of the uppercase $\mathrm{X}$ and lowercase $\mathrm{x}$ in Equations 6 and 7.
>
> **A3**: In Equations 6 and 7, $\mathrm{X}$ represents the complete source sentence, while $\mathrm{x}$ denotes the received source prefix. This distinction is noted near Equation 3 (around lines 236-240) and Equation 6 (around line 279). We will further clarify this point in our revised paper.
>
> **Q4**: Explanation of the ablation studies.
>
> **A4**: The numerical results of the ablation studies in our paper are presented as follows. The "$\Delta$ BLEU" means the improvement in SacreBLEU score of SimulPL compared to MSFT.
>
> |        | Metrics       | n=3   | n=5   | n=7   | n=10  | n=15  | n=20  | Average |
> | ------ | ------------- | ----- | ----- | ----- | ----- | ----- | ----- | ------- |
> | MSFT   | LAAL          | 1.77  | 2.95  | 4.11  | 5.90  | 9.13  | 12.02 | 5.98    |
> |        | BLEU          | 21.80 | 24.04 | 25.96 | 27.06 | 28.92 | 29.38 | 26.19   |
> | SimulPL | LAAL          | 1.73  | 2.90  | 4.04  | 5.87  | 9.08  | 12.01 | 5.94    |
> |        | BLEU          | 22.51 | 24.94 | 26.76 | 27.97 | 29.94 | 30.05 | 27.03   |
> |        | $\Delta$ BLEU | +0.71 | +0.90 | +0.80 | +0.91 | +1.02 | +0.67 | +0.83   |
>
> The results show that SimulPL outperforms MSFT at all latency levels, with a maximum increase of 1.02 BLEU and an average increase of 0.83 BLEU, which is a notable improvement. Furthermore, based on analyses from existing work [1], the gains from incorporating SimulDPO into SimulPL are also reasonable. Therefore, we believe that SimulPL indeed outperforms MSFT, and concerns about its effectiveness are unnecessary.
>
> **Reference**
>
> [1] Saeidi, Amir, Shivanshu Verma, and Chitta Baral. "Insights into Alignment: Evaluating DPO and its Variants Across Multiple Tasks." *arXiv preprint arXiv:2404.14723* (2024).

---

> ### Author Response · Authors · 2024-11-20
> **Thanks for Reviewer TojP's valuable comments! Here is the response to Reviewer TojP (Part II)**
>
> **Once again, thank you for your valuable feedback! We hope our responses have addressed your concerns and questions. If so, we would greatly appreciate it if you could consider acknowledging our efforts and kindly increasing your score. Should you have any further questions, please feel free to reach out. We look forward to further discussions with you!**

---

> > ### Comment · Reviewer_TojP · 2024-11-26
> >
> > The authors have addressed some of my doubts, so I have increased my score.

---

> > > ### Author Response · Authors · 2024-11-27
> > > **Thanks for Further Feedback!**
> > >
> > > Thank you very much for recognizing our efforts and for the increase in your score. We truly appreciate your feedback and will take your suggestions into account for further revisions in the final version.

---

### Official Review · Reviewer_kP65 · 2024-11-04

**Soundness:** 3
**Presentation:** 3
**Contribution:** 3
**Rating:** 8
**Confidence:** 4

**Summary:**

This works present a preference learning framework for simultaneous tasks like SiMT. Particularly, the proposed SimulPL framework categorizes SiMT human preferences into five aspects: translation quality preference, monotonicity preference, key point preference, simplicity preference, and latency preference.

**Strengths:**

1. This is an intriguing question and a useful framework for practical SiMT scenarios. While SiMT has been extensively studied in the research community, most papers have focused on automatic metrics rather than real human preferences. This work proposed 5 human preferences that are closely related to the end-users of SiMT.

2. The proposed optimization objective aligns well with the proposed human preferences. For example, the output length constraint is applied for latency preference. The derived final objective in Eq(9) is further to improve its read/write policy.

**Weaknesses:**

1. The data construction relies on GPT-4/4o. What is the cost of construction the training data? Does it mean GPT-4/4o is the upper bound of human preference alignment? Or this work is to GPT-4 preference alignment?  Figure 2 only shows the win-rate between GPT-4 translation and original one. Is there a comparison between GPT-4 and professional interpreter?

2. In the experiments, you only tune the hyper-parameter $\alpha$, and set $c_t$ as 0.5 to control the read/write. In general, tuning the thresholds of $c_t$ may introduce various read/write strategies. Did you try other threshold?

**Questions:**

1. Since real-time translation is a key requirement of SiMT, LLM-based SiMT may need to re-translate each partial source sentence, as the decoder-only architecture of LLMs does not permit the insertion of additional source tokens without disrupting the positional indexes of the already translated target tokens. I want the authors to justify this issue.

2. Continued on Weaknesses 1, given some GPT-4/4o translations, is that possible to ask the professional interpreter to select the high quality translation? Then a classifier or reward model can be implemented to filter out the translations align most with human.

---

> ### Author Response · Authors · 2024-11-20
> **Thanks for Reviewer kP65's valuable comments! Here is the response to Reviewer kP65 (Part I)**
>
> Thank you for recognizing our work and providing valuable feedback—it is truly encouraging! The following are our detailed responses.
>
> **Q1:** Regarding data construction.
>
> **A1:** **Training Cost:** The cost of annotating all the data was approximately \\$960. Given the recent price reduction of GPT-4o from \\$5 to \\$2.5, as well as the introduction of GPT-4o mini, this cost can be further reduced.
>
> **Data Quality:** *First*, our test set was manually revised by professional interpreters. This is mentioned in lines 221-222 on Page 5 of our paper. Therefore, the evaluation results based on these test sets strongly support the effectiveness of SimulPL in aligning with human preferences rather than GPT-generated preferences.
>
> *Second*, as you suggested, we compare the GPT-generated translations with the manually revised translations. Please refer to our **General Response: Further Evaluation of GPT-generated Data** for more details.
>
> **Q2:** The impact of the confidence threshold.
>
> **A2:** Thank you for your advice. As you suggested, we validate the impact of the confidence threshold on SimulPL. Specifically, we assess the performenace of SimulPL on  $\gamma=0.1,0.3,0.5,0.7,0.9$ with different $n$ during inference. Due to time constraints, we only perform the experiments on Zh$\rightarrow$En task. The results are as follows:
>
> |      | $\gamma$=0.1 |           | $\gamma$=0.3 |           | $\gamma$=0.5 |           | $\gamma$=0.7 |           | $\gamma$=0.9 |           |
> | ---- | ------------ | --------- | ------------ | --------- | ------------ | --------- | ------------ | --------- | ------------ | --------- |
> | $n$  | LAAL         | SacreBLEU | LAAL         | SacreBLEU | LAAL         | SacreBLEU | LAAL         | SacreBLEU | LAAL         | SacreBLEU |
> | 3    | 2.03         | 20.76     | 1.99         | 21.51     | 1.73         | 22.51     | 1.99         | 22.36     | 2.04         | 22.51     |
> | 5    | 3.21         | 24.25     | 3.19         | 24.89     | 2.90         | 24.94     | 3.11         | 25.24     | 3.29         | 24.97     |
> | 7    | 4.36         | 26.48     | 4.36         | 26.58     | 4.04         | 26.76     | 4.28         | 26.94     | 4.45         | 26.78     |
> | 10   | 6.27         | 28.17     | 6.25         | 28.10     | 5.87         | 27.97     | 6.18         | 28.45     | 6.36         | 27.81     |
> | 15   | 9.42         | 29.86     | 9.40         | 29.80     | 9.08         | 29.94     | 9.22         | 30.13     | 9.46         | 30.20     |
> | 20   | 12.37        | 30.17     | 12.35        | 30.20     | 12.01        | 30.05     | 12.15        | 30.52     | 12.46        | 30.47     |
>
> When $\gamma$ is set to a small value ($\gamma=0.1$), the model is allowed to output tokens with low confidence, which results in a decline in translation quality, especially in low latency levels (LAAL < 4). When $\gamma$ is set to a higher value ($\gamma=0.9$), the model imposes stricter constraints on token quality, introducing unnecessary delays. We plan to further explore the impact of $\gamma$ in our future work.

---

> ### Author Response · Authors · 2024-11-20
> **Thanks for Reviewer kP65's valuable comments! Here is the response to Reviewer kP65 (Part II)**
>
> **Q3:** The requirement for recalculating in LLMs.
>
> **A3:** As you pointed out, existing LLM-based methods [1] [2] [3] often require recalculating the hidden states after the insertion point when new source tokens are input. However, previous work [3] has shown that this recalculation does not significantly affect real-world performance. Furthermore, techniques such as SimulMask [4] can also be applied to existing LLM-based SiMT models to mitigate the overhead of recomputing hidden states.
>
> Additionally, our work focuses on improving the SiMT human preferences alignment. Therefore, we adopt a simpler LLM-based SiMT model, which does not impact the applicability of SimulPL to LLM-based SiMT models that do not require repeated hidden state calculations.
>
> Thank you for your insightful question. We will explore this issue further in our future work.
>
> **Reference**
>
> [1] Wang, Minghan, et al. "Simultaneous machine translation with large language models." *arXiv preprint arXiv:2309.06706* (2023).
>
> [2] Koshkin, Roman, Katsuhito Sudoh, and Satoshi Nakamura. "Transllama: Llm-based simultaneous translation system." *arXiv preprint arXiv:2402.04636* (2024).
>
> [3] Guo, Shoutao, et al. "SiLLM: Large Language Models for Simultaneous Machine Translation." *arXiv preprint arXiv:2402.13036* (2024).
>
> [4] Raffel, Matthew, Victor Agostinelli, and Lizhong Chen. "Simultaneous Masking, Not Prompting Optimization: A Paradigm Shift in Fine-tuning LLMs for Simultaneous Translation." *arXiv preprint arXiv:2405.10443* (2024).
>
> **Q4:** Feasibility of constructing a reward model.
>
> **A4:** Thank you for your insightful suggestion. The idea of constructing a reward model sounds promising and could potentially offer higher performance ceilings. We will explore this direction further in future work.
>
> **Once again, we deeply appreciate your recognition of our work and your valuable suggestions! We are committed to incorporating your suggestions into our future work.**

---

> > ### Comment · Reviewer_kP65 · 2024-11-27
> >
> > Thanks for the detailed explanation. I increased my score.

---

> > > ### Author Response · Authors · 2024-11-28
> > > **Thanks for Further Feedback!**
> > >
> > > Thank you very much for recognizing our work and for increasing our score! We will carefully consider your suggestions in the revised version.

---

### Author Response · Authors · 2024-11-20
**General Response: Further Evaluation of GPT-generated Data. (Part II)**

**For human evaluation**, following Kocmi et al. [1] and Xu et al. [2], we randomly sample 200 sentences from the test set and employ professional interpreters, who are not involved in the revision process, to manually assess their GPT-generated translations and manually-revised translations. Our evaluation criteria are as follows:

- **0**: The translation is poor and fails to convey any meaningful information.
- **2**: The translation conveys some of the speaker’s information but misses key points. It also includes unnecessary reordering, leading to unnecessary delays in real-time scenarios, and uses less common expressions that make it harder for the audience to quickly understand.
- **4**: The translation conveys all the important information with minimal unnecessary reordering. It uses simple expressions that generally meet the audience's needs, though there are still some minor issues.
- **6**: The translation is a perfect simultaneous interpretation, accurately conveying the speaker's key points while omitting unnecessary details. It uses expressions that align with spoken language conventions, significantly reflecting human preferences in simultaneous interpretation.

Due to time constraints, we conduct this evaluation only on the Zh$\rightarrow$En test set. We calculate their average scores, win-tie-loss rates, and the distribution of their scores. The results are as follows:

| References       | Average Score | Win-Tie-Lose |            |           | Distribution of Scores |       |        |        |
| ---------------- | ------------- | ------------ | ---------- | --------- | ---------------------- | ----- | ------ | ------ |
|                  |               | win ratio    | lose ratio | tie ratio | 0                      | 2     | 4      | 6      |
| GPT-generated    | 5.37          | 6.00%        | 12.00%     | 82.00%    | 0.50%                  | 5.50% | 19.00% | 75.00% |
| Manually Revised | 5.57          | 12.00%       | 6.00%      | 82.00%    | 0.00%                  | 1.00% | 19.50% | 79.50% |

As shown in the table, the scores of GPT-generated translations are very close to those are manually revised, with only a 12% loss ratio. The score distribution shows that 94% of the GPT-generated translations scored 4 or higher. These results suggest that the GPT-generated data aligns well with SiMT human preferences.

**Reference**

[1] Tom Kocmi, Rachel Bawden, Ondřej Bojar, Anton Dvorkovich, Christian Federmann, Mark Fishel, Thamme Gowda, Yvette Graham, Roman Grundkiewicz, Barry Haddow, Rebecca Knowles, Philipp Koehn, Christof Monz, Makoto Morishita, Masaaki Nagata, Toshiaki Nakazawa, Michal Novák, Martin Popel, and Maja Popović. 2022. Findings of the 2022 Conference on Machine Translation (WMT22). In *Proceedings of the Seventh Conference on Machine Translation (WMT)*, pages 1–45, Abu Dhabi, United Arab Emirates (Hybrid). Association for Computational Linguistics.

[2] Xu, Haoran, et al. "Contrastive Preference Optimization: Pushing the Boundaries of LLM Performance in Machine Translation." *Forty-first International Conference on Machine Learning*.

---

### Author Response · Authors · 2024-11-20
**General Response: Further Evaluation of GPT-generated Data. (Part I)**

The references in our test sets are manually revised by professional interpreters, which is mentioned in lines 221-222 on Page 5 of our paper. To further validate the quality of our constructed dataset, we add an analysis of GPT-generated translations and those manually revised by interpreters on the test set. The experiments include both automated evaluation and human evaluation.

**For automated evaluation**, we compare the multi-aspect evaluation results of GPT-generated translations and the manually revised translations. To facilitate comparison, we also provide the multi-aspect evaluation results for the original references of the test set. These results are shown as follows:

| references       | Zh$\rightarrow$En   |                 |                  |                | De$\rightarrow$En   |                 |                  |                | En$\rightarrow$Zh   |                 |                  |                |
| ---------------- | ------------------- | --------------- | ---------------- | -------------- | ------------------- | --------------- | ---------------- | -------------- | ------------------- | --------------- | ---------------- | -------------- |
|                  | NIR(%) $\downarrow$ | DD $\downarrow$ | SLR $\downarrow$ | Ref-free COMET | NIR(%) $\downarrow$ | DD $\downarrow$ | SLR $\downarrow$ | Ref-free COMET | NIR(%) $\downarrow$ | DD $\downarrow$ | SLR $\downarrow$ | Ref-free COMET |
| GPT-generated    | 6.48                | 6.55            | 0.49             | 78.32          | 5.49                | 5.71            | 0.92             | 80.86          | 10.65               | 5.63            | 0.88             | 81.24          |
| Manually Revised | 6.71                | 6.69            | 0.52             | 79.05          | 5.52                | 5.84            | 0.95             | 81.89          | 10.47               | 5.63            | 0.88             | 81.42          |
| Origin           | 13.21               | 7.53            | 0.74             | 79.05          | 7.80                | 6.29            | 1.09             | 80.78          | 12.89               | 6.19            | 0.89             | 75.77          |

As shown in the table, the translations generated by GPT-4/4o are either clearly superior to or comparable with the original references in terms of monotonicity, key points, simplicity, and translation quality. Moreover, these results are very close to the manually revised translations. This indicates that the quality of the GPT-generated data is both reliable and aligned well with human preferences.

---

### Author Response · Authors · 2024-11-26
**General Response to Reviewer Feedback and Updates to Our Paper.**

We would like to thank all the reviewers for their valuable feedback!

We are pleased to hear that the reviewers found the study of human preferences in SiMT scenarios **a fascinating topic** (Reviewer kP65, Reviewer 1TVX); that **our SimulPL framework is considered practical for SiMT scenarios** (Reviewer kP65, Reviewer TojP); that **our categorized five aspects of human preferences  for SiMT tasks resonate with users** (Reviewer kP65, Reviewer qWwc); that **our experimental results are comprehensive and convincing** (Reviewer TojP, Reviewer qWwc, Reviewer 1TVX); and that the paper is **clear and easy to understand** (Reviewer qWwc).

We have addressed all the reviewers' comments and revised the paper accordingly. The revised version primarily includes the following significant updates, with modified sections marked in red.

1. In the Introduction, we **clarified our contributions more clearly** (Reviewer TojP).
2. We have further clarified **the distinction between $\mathrm{X}$ and  $\mathrm{x}$  in Equations 6 and 7** (Reviewer TojP).
3. In Section 5.6, we **discussed the generalization of SimulPL to other tasks** (Reviewer qWwc).
4. In Appendix A.2, we **added further evaluation of GPT-generated data** (Reviewer kP65, Reviewer TojP, Reviewer qWwc).
5. In Appendix C, we **explained the difference between our length-based constraints for latency preference and existing methods** (Reviewer qWwc).
6. In Appendix D, we **analyzed the impact of the confidence threshold on SimulPL’s performance during inference** (Reviewer kP65, Reviewer qWwc).
7. In Appendix G, we **provided additional analysis experiments for De$\rightarrow$En task** (Reviewer qWwc).

We hope that our revisions address the reviewers' concerns. If so, we sincerely hope that the reviewers might consider raising the scores in light of the updates. We promise to further improve the paper in our final revision.

**Finally, we sincerely thank all the reviewers, ACs, and PCs for their time and effort!**

---

### Meta-Review · Area_Chair_dXTE · 2024-12-19

**Metareview:**

The paper introduces a framework for simultaneous machine translation (SiMT) that aligns with human preferences, categorized into five aspects: translation quality, monotonicity, key points, simplicity, and latency.

The paper's strengths include its comprehensive approach to addressing human preferences in SiMT, the innovative use of GPT-4/4o for generating preference data, and the convincing experimental results demonstrating significant improvements in alignment with human preferences. The paper is well-written and easy to understand, making complex ideas accessible to the reader.

The weaknesses include the reliance on GPT-4/4o for data construction, raising questions about the cost and quality of the generated data, and the lack of detailed comparisons with professional interpreters. The presentation could be improved, particularly in sections discussing the empirical validation and the logical flow between different parts of the paper.

All the reviewers are positive about the paper. I recommend accepting it.

**Additional Comments On Reviewer Discussion:**

The authors well addressed the reviewers' concerns.

---

### Decision · Program_Chairs · 2025-01-22

Accept (Poster)